# Nuclear transporter Importin-13 plays a key role in the oxidative stress transcriptional response

K. A. Gajewska[1], H. Lescesen[1], M. Ramialison[2], K. M. Wagstaff[1] & D. A. Jans [1✉]

The importin superfamily member Importin-13 is a bidirectional nuclear transporter. To delineate its functional roles, we performed transcriptomic analysis on wild-type and Importin-13-knockout mouse embryonic stem cells, revealing enrichment of differentially expressed genes involved in stress responses and apoptosis regulation. De novo promoter motif analysis on 277 Importin-13-dependent genes responsive to oxidative stress revealed an enrichment of motifs aligned to consensus sites for the transcription factors specificity protein 1, SP1, or Kruppel like factor 4, KLF4. Analysis of embryonic stem cells subjected to oxidative stress revealed that Importin-13-knockout cells were more resistant, with knockdown of SP1 or KLF4 helping protect wild-type embryonic stem cells against stress-induced death. Importin-13 was revealed to bind to SP1 and KLF4 in a cellular context, with a key role in oxidative stress-dependent nuclear export of both transcription factors. The results are integral to understanding stress biology, highlighting the importance of Importin-13 in the stress response.

[1] Biomedicine Discovery Institute, Monash University, Clayton, VIC, Australia. [2] Australian Regenerative Medicine Institute and Systems Biology Institute, Monash University, Clayton, VIC, Australia. ✉email: david.jans@monash.edu

Subcellular compartmentalisation of signalling proteins, including transcription factors (TFs), is central to gene regulation that coordinates a range of cellular processes such as signal transduction, cell proliferation, differentiation, and development. Signalling molecules, such as TFs, larger than ~40 kDa are conventionally transported between the nuclear and cytoplasmic compartment by members of the Importin (IPO) superfamily of transport receptors[1,2]. Importin-13 (IPO13) is one of only three mammalian family members capable of mediating both nuclear import and export of specific protein cargoes[3–5] including TFs[6] that play key roles in brain[7,8], testis[9] and fetal lung development[10,11].

A number of processes are critical to normal organogenesis and limb development in mammals, including specific and highly organised cell proliferation, differentiation, and apoptosis. The redox environment of the cells within the developing embryo controls cell fate[12], whereby stem cell proliferation is promoted in reducing environments with low to mild levels of Reactive Oxygen Species (ROS), and inhibited under conditions of higher ROS levels, which favour pro-apoptotic signalling[13,14]. ROS are generated by the incomplete reduction of oxygen, and include derivative species such as hydrogen peroxide ($H_2O_2$), superoxide anions ($O_2^-$), and hydroxyl radicals ($HO^{\bullet}$)[15,16]. ROS are generated as by-products of normal biological processes within the eukaryotic cell, such as aerobic metabolism, and can act as second messengers in various signal transduction pathways[16–18]. They can also be taken up from exogenous sources or produced in response to environmental stress. While transient fluctuations are normally managed by the cellular antioxidant defences, sustained exposure to ROS can result in oxidative stress, with the potential to damage lipids, protein, and DNA[18]. The cellular response to oxidative stress varies from growth arrest to senescence, to cell death by apoptosis and necrosis[19]. Although cellular responses such as the response to oxidative stress are characterised by changes in the nucleo-cytoplasmic distribution of signalling molecules and TFs, classical IPO-dependent nuclear transport pathways are known to be inhibited by oxidative stress, as well as heat shock and osmotic stress, in part, through key nuclear transport factors becoming mislocalised[20–25].

This study examines IPO13's contribution to cell signalling pathways through transcriptomic profiling, demonstrating that the altered expression of a number of oxidative stress-responsive genes in response to oxidative stress is dependent on IPO13. We show that this set of genes is enriched in consensus sites for key TFs, including the oxidative stress-inducible TF specificity protein 1 (SP1), a regulator of cell growth and development that can act as a cofactor in p53-mediated pro-apoptotic transcriptional repression[26,27], and Kruppel like factor 4 (KLF4)[28,29], a zinc finger containing transcription factor with roles in proliferation[30], differentiation[31], cell cycle arrest following DNA damage[32] and apoptosis. Significantly in this context, we find that knock out of IPO13 protects mouse embryonic stem cells (mESCs) from oxidative stress-induced death, with siRNA knockdown of either SP1 or KLF4 in IPO13$^{+/+}$ also able to confer resistance to $H_2O_2$. We further show that IPO13 interacts with and regulates the nuclear export of both KLF4 and SP1 under conditions of oxidative stress, this transport function of IPO13 with respect to KLF4 and SP1 in response to oxidative stress likely contributing to modulation of cell survival under these conditions. To the best of our knowledge, our findings represent the first documented examples of IPOβ family-dependent nuclear transport of specific cargoes in stress, establishing a key role for IPO13 in stress responses through its ability to transport key factors in stress conditions.

## Results

**The IPO13-dependent oxidative stress transcriptome.** IPO13 mediates bidirectional nuclear transport of various important cargoes and TFs[6,33]. To shed light on its biological role, we subjected a previously characterised IPO13 gene knockout model in mESCs[4] to whole-transcriptome analysis. Next Generation RNA-Sequencing (RNA-seq) followed by differential gene expression analysis revealed a total of 338 genes that were significantly downregulated and 536 genes that were significantly up-regulated in IPO13$^{-/-}$ compared to IPO13$^{+/+}$ mESCs (Sup. Fig. 1a; Sup. Data 1). Gene ontology analysis of the genes with annotated function (651 in all) using the David Bioinformatics Functional Annotation Tool[34,35] revealed the greatest enrichment of stress response and apoptotic pathways including DNA damage response, heat shock, chemical stress, and intrinsic apoptosis signalling respectively (Sup. Fig. 1b).

To explore the role of IPO13 in the cellular stress response, we performed RNA-seq on IPO13$^{+/+}$ and IPO13$^{-/-}$ ESCs treated without or with a sublethal concentration of $H_2O_2$ for 1 h to induce oxidative stress followed by a recovery period of 2 h prior to total RNA extraction for RNA sequencing and bioinformatic analysis (Fig. 1a–d). As one of the key metabolites in oxidative stress, $H_2O_2$ is a by-product of normal cellular metabolism that has been used extensively in mammalian cell models to induce oxidative stress[36–38]. We confirmed that the sublethal $H_2O_2$ treatment used here could induce oxidative stress in the ESCs by flow cytometric analysis of $H_2O_2$ treated IPO13$^{+/+}$ and IPO13$^{-/-}$ cells stained with CellROX Green, a probe that is stably fluorescent when oxidised by ROS. A significant increase in relative fluorescence intensity was observed after treatment with 125 μM $H_2O_2$ for 1 h in both the IPO13$^{+/+}$ and IPO13$^{-/-}$ ESCs (Fig. 1e–f), indicating that treatment of ESCs with $H_2O_2$ could upregulate cellular ROS levels. ROS levels returned to basal in both IPO13$^{+/+}$ and IPO13$^{-/-}$ cells after removal of the $H_2O_2$ treatment and 2 h recovery time after the stress treatment.

Across the four conditions (IPO13$^{+/+}$ and IPO13$^{-/-}$ with or without oxidative stress), a total of 2352 genes were found to be ≥ log2-fold differentially expressed (FDR cut off of 0.05; Fig. 1a), the multi-dimensional scaling plot (Fig. 1b) highlights the significant differences between IPO13$^{+/+}$ and IPO13$^{-/-}$ ESCs, as well as between the cell lines in the presence and absence of oxidative stress. Comparison of the specific gene subsets responsive to oxidative stress (Sup. Data 2 and 3) identified 277 IPO13-dependent genes up- or down-regulated exclusively in IPO13$^{+/+}$ and not in IPO13$^{-/-}$ ESCs (Fig. 1c, d). Gene ontology analysis of this "IPO13-dependent oxidative stress transcriptome" revealed enrichment of stress response pathways, including the unfolded protein response and apoptosis (Fig. 2a), as did an analysis of the 300-gene subset exclusively differentially expressed in response to oxidative stress in the IPO13$^{-/-}$ ESCs (Fig. 2b). Stress response pathways again appeared to be highly enriched, including pathways relating to apoptotic cell death.

A subset of the differentially regulated genes from the 277-gene IPO13-dependent oxidative stress transcriptome, covering a range of extents of differential expression as well as representing genes annotated to have a role in stress response, were selected for validation by real-time quantitative PCR (qPCR). The *Cyp2s1* gene, downregulated in response to oxidative stress independent of the presence of IPO13 (see Sup. Data 2 and 3), was included as a control. Results were universally consistent with the RNA-seq data, (Sup. Fig. 2), with oxidative stress-inducing expression changes in all of the genes apart from *Cyp2s1* in IPO13$^{+/+}$ but not IPO13$^{-/-}$ ESCs, confirming the authenticity of the differential expression of genes within the Next Generation Sequencing identified IPO13 oxidative stress transcriptome.

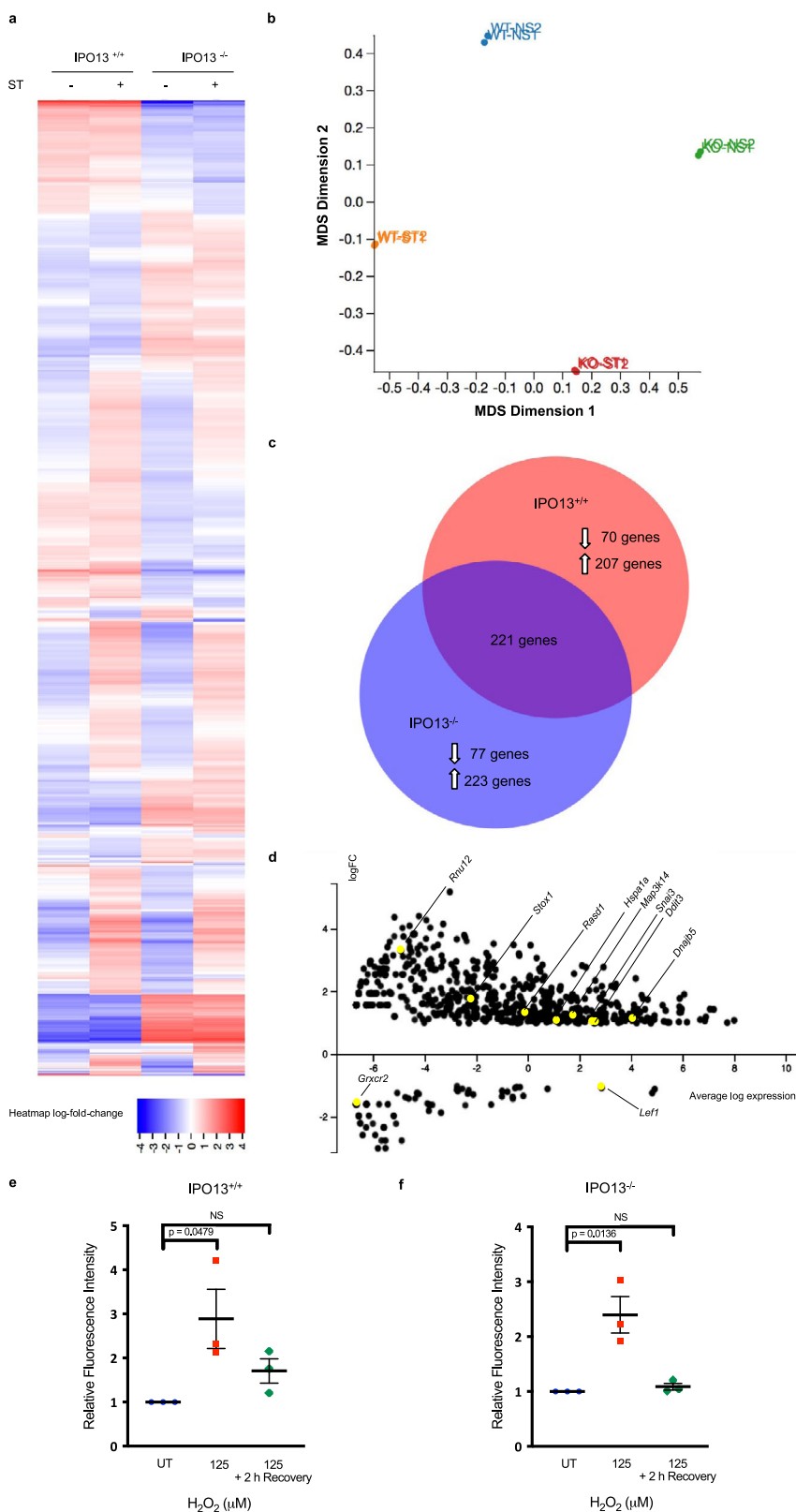

Taken together, IPO13 appears to be a key factor in the transcriptional response to oxidative stress.

**The IPO13 regulated transcriptional network**. Since IPO13's primary function is to facilitate nucleocytoplasmic shuttling of specific cargo proteins, we hypothesised that the effects of IPO13 knock-out on the gene expression profile of the ESCs in both untreated and oxidative stress treated conditions are likely mediated through altered transport of transcriptional activators and/or repressors in the IPO13 knock-out cells compared to wild type. In order to unravel the IPO13-dependent transcriptional regulatory network under stress, we set out to identify the TFs responsible for the differential expression of the 277 genes of the IPO13 oxidative stress transcriptome.

**Fig. 1 The IPO13-dependent transcriptome in oxidative stress. a** Heat-map of transcript data for IPO13$^{+/+}$ and IPO13$^{-/-}$ mESCs with or without treatment with 125 μM hydrogen peroxide (H$_2$O$_2$) for 1 h followed by 2 h recovery (ST; stress treated). Log$_2$-fold differences in expression across the four data sets (FDR cut-off = 0.05) are shown for 2352 gene models (horizontal axis). **b** Multi-dimensional scaling (MDS) plot summarizing gene expression profiles of IPO13$^{+/+}$ ("WT") and IPO13$^{-/-}$ ("KO") ESCs for control (NS; no stress) and oxidative stress-treated (ST) conditions. **c** Venn diagram for the differentially expressed genes in response to stress in IPO13$^{+/+}$ and IPO13$^{-/-}$ ESCs (498 and 521 genes, respectively). **d** MA plot showing differentially expressed genes with a log$_2$-fold change increase or decrease in expression (FDR cut-off = 0.05) in response to oxidative stress compared to untreated ESCs. Log fold change (logFC) values are plotted against log expression values (standardized read counts). Genes highlighted in yellow were selected for validation. **e–f** IPO13$^{+/+}$ and IPO13$^{-/-}$ ESCs were treated with 125 μM H$_2$O$_2$ as indicated for 1 h and then stained for 30 min with 5 μM CellROX Green either immediately, or after a 2 h recovery period, and then analysed by flow cytometry. Quantification of the median fluorescence intensity relative to the untreated (UT) sample are shown representing the mean ± SEM ($n = 3$ independent experiments); $p$ values (two-tailed student's $t$-test) top to bottom left to right: $p = 0.0617$, $p = 0.0479$, $p = 0.2167$, and $p = 0.0136$. Source data are provided as a Source Data file.

Toward this end, we performed DNA motif enrichment analysis (Fig. 3a) to identify consensus TF binding motifs that were enriched in the regulatory regions of IPO13-dependent oxidative stress-related genes. Enriched trimethylation (H3K4me3) or monomethylation of histone H3 at lysine 4 (H3K4me1) and acetylation of histone H3 lysine 27 (H3k27ac) are chromatin signatures of promoters and enhancers in mammalian genomes respectively[39]. Genomic regions for our genes of interest in mESCs for H3K4me3, H3K4me1, and H3k27ac were accessed from ChIP-seq data in mouse ENCODE[40], and subjected to analysis using three independent motif discovery tools (MEME-ChIP[41], RSAT-Peak Motifs[42,43] and Trawler[44,45]). Motifs discovered by multiple tools were given weight and aligned to known TF binding motifs by STAMP[46] using the JASPAR database[47]. Motifs and the corresponding TF were then matched to the genes in which they were found.

Four TFs and their corresponding motifs (Fig. 3b) were selected for validation; Nuclear Factor of Activated T-cells 2 (NFATC2, NFAT1), Specificity Protein 1 (SP1), Ras Responsive Element Binding Protein 1 (RREB-1), and Kruppel like factor 4 (KLF4). These were selected firstly based on the relative measure of similarity (p-value) of the candidate motif to the transcription factor-binding motif as ranked by STAMP; NFAT1 (E value: 3.1860e$^{-05}$), RREB-1 (E value: 3.2009e$^{-11}$), KLF4 (E value: 3.3132e$^{-10}$) and SP1, which was aligned to both the KLF4 aligned motif (3.8086e$^{-10}$) and the RREB-1 aligned motif (4.3809e-13). Secondly, published NFAT1[48], KLF4[49], and SP1[50] ChIP-seq datasets contained peak sequences that corresponded to the most genes within the 277 gene subset, verifying that these TFs indeed bind to these genes. Finally, a literature review of the biological functions of each TF revealed roles highlighted by the functional annotation of our subset of genes (Fig. 2a), including regulation of apoptotic pathways. Specifically, NFAT1[51–53] and RREB-1[54] contribute to pro-apoptotic pathways, whilst SP1[27,55–57] and KLF4[29,58–60] have been reported to play both pro-apoptotic and pro-survival roles. Importantly, these four TFs appear to regulate the expression of a large portion of the network (135 genes) of the 277 IPO13-dependent oxidative stress-responsive genes (Fig. 3c).

To begin to validate the contribution of these TFs to the differential expression seen in the IPO13 oxidative stress transcriptome, we performed RT-qPCR analysis on RNA extracted from IPO13$^{+/+}$ ESCs subjected to siRNA-induced knockdown for each TF. A set of genes containing the consensus sites for NFAT1, SP1, RREB-1, and/or KLF4 (see Fig. 3c) were tested, in addition to the control Cyp2s1. We initially determined that, in the case of the Snai3 gene, of all the genes tested, knockdown of KLF4 or RREB-1 in unstressed cells could significantly upregulate mRNA levels, likely an indication of KLF4 and RREB-1 playing a role in Snai3 basal expression as well as in response to stress. To correct for this, we relativised all results to the respective basal levels without stress; analysis revealed that knockdown of at least one of the four TFs abrogated

the transcriptional response to oxidative stress for the majority of the selected genes, resulting in a lack of response of IPO13$^{+/+}$ ESCs that resembled that of IPO13$^{-/-}$ ESCs. Results for the Cyps21 control showed that response to stress was unaffected by the specific knockdown, underlining the specificity of the effects for the other genes (Sup. Fig. 3). In the case of stress-induced up-regulation of Snai3, knockdown of any one of KLF4, SP1 or RREB-1 was sufficient to silence oxidative stress-responsive upregulation (Sup. Fig. 3b, c, d). Analogously, siRNA for KLF4, SP1, or RREB-1 (Sup. Fig. 3b, c, d) could abrogate stress-induced downregulation of the Lef1 gene. Other genes showing marked, but not statistically significant, reductions in response to stress after siRNA knockdown of specific TFs included Dok1 where siRNA to either SP1 or RREB-1 decreased responsiveness (Sup. Fig. 3c, d). The clear implication here is that multiple TFs, sometimes in combination, play key roles in the IPO13-dependent transcriptional response to stress. Significantly, previous studies have reported synergistic actions of KLF4 and SP1 in gene regulation[61], while NFAT1 and SP1 have also been shown to functionally interact to regulate gene expression[62]. Several genes validated here by qPCR as targets of at least one of the four TFs of interest (Sup. Fig. 3) have been reported to regulate redox homeostasis and cell death[63–69]. This includes Stox1, reported as a master regulator of redox balance in mouse placentas and trophoblast cells[63] and is associated with apoptosis downstream of the PI3K/Akt signalling pathway whereby down-regulation of Stox1 in trophoblasts could promote apoptosis via inhibition of the PI3K/Akt pathway[64]. Ppard is believed to regulate ROS homeostasis[65,66] and Il5 has been reported as both a pro-apoptotic factor and inhibitor of cell death in varying cell types[67,68]. Finally, Rasd1 has been reported to induce apoptosis when transfected into lung adenocarcinoma cells[69].

**SP1 or KLF4 knockdown contribute to IPO13-dependent cell death.** The ability of cells to respond to stress is dependent upon the activation of numerous signalling pathways that regulate both adaption/resistance to stress[19] and cell death where irreparable damage has occurred[70]. To investigate the role of the IPO13-dependent transcriptional response to potentially lethal levels of stress, we used PI staining/flow cytometry to assess IPO13$^{+/+}$ and IPO13$^{-/-}$ cell viability either immediately (no recovery) or 2 h after oxidative stress over a range of H$_2$O$_2$ concentrations (Fig. 4a, b).

Across the range of treatments, IPO13$^{-/-}$ ESCs were consistently more viable than IPO13$^{+/+}$ ESCs, with IPO13 knock out significantly reducing the percentage of H$_2$O$_2$-induced cell death compared to the IPO13$^{+/+}$ ESCs (Fig. 4a), implying that the absence of IPO13 may contribute resistance to H$_2$O$_2$. After 2 h recovery, the levels of cell death were lower in IPO13$^{+/+}$, but not IPO13$^{-/-}$ ESCs, with the percentage of cell death in IPO13$^{+/+}$ ESCs treated with 300 or 600 μM H$_2$O$_2$, reduced by 18 and 24% respectively (Fig. 4a, b). The implication is that while IPO13

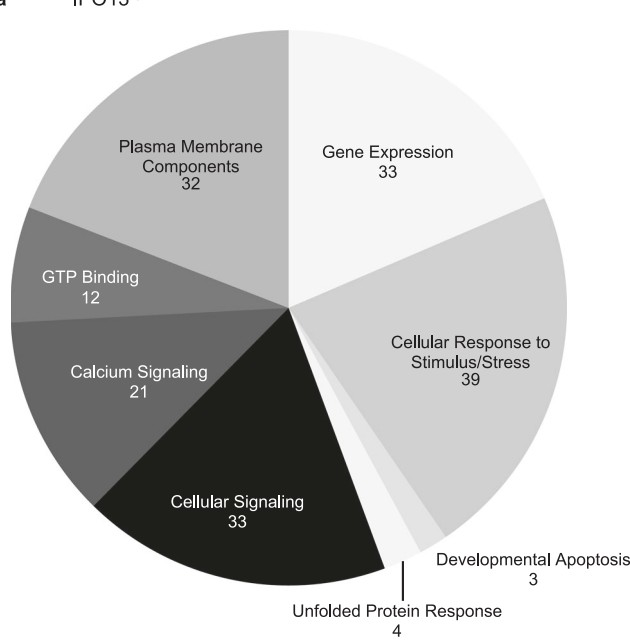

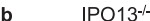

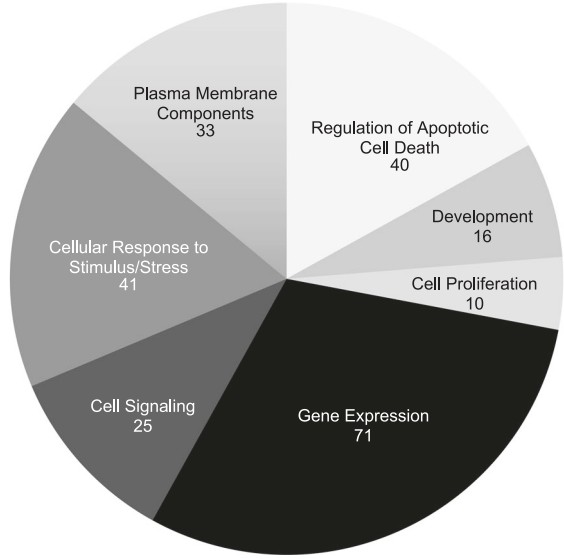

**Fig. 2 Gene ontology analysis of differentially regulated genes in response to oxidative stress in ESCs.** Pathways analysis was performed on differentially regulated genes responsive to oxidative stress in (**a**) IPO13$^{+/+}$ or (**b**) IPO13$^{-/-}$ ESCs using DAVID Bioinformatics Resources 6.8, using the Functional Annotation Tool for the 178 and 236 annotated genes, respectively.

appears to sensitise to oxidative stress-induced cell death, it may also contribute to repair in the post-stress recovery phase, implying that IPO13 may have roles in promoting both survival and death depending on the intensity and duration of exposure to oxidative stress. To confirm formally the key importance of IPO13 in this context, we tested whether ectopic expression of GFP-tagged IPO13 in the IPO13$^{-/-}$ ESCs could confer increased sensitivity to H$_2$O$_2$. IPO13 expressing IPO13$^{-/-}$ cells were significantly more sensitive than IPO13$^{-/-}$ cells expressing GFP

alone after treatment with H$_2$O$_2$. (Sup. Fig. 4), consistent with the functional rescue of IPO13's role in this context.

The contribution of our candidate TFs to IPO13-dependent oxidative stress-induced cell death was tested by subjecting IPO13$^{+/+}$ ESCs with siRNA-induced knockdown of SP1 or KLF4 to the same stress treatment protocols (Fig. 4a, b). Results were unequivocal in that knockdown of either TF significantly decreased cell death in response to 1 h H$_2$O$_2$ stress, bringing the cell death to a level comparable to that of IPO13$^{-/-}$. Consequently, KLF4 and SP1 may be involved in the mechanism by which IPO13 regulates cell survival in the oxidative stress response. Given that the central function of IPO13 is the nuclear transport of proteins, KLF4 and SP1 are most likely transported by IPO13 under conditions of oxidative stress to affect cell survival, presumably, through transcriptional regulation.

To confirm that the observations with respect to resistance to oxidative stress were not limited to H$_2$O$_2$-induced oxidative stress, cells were treated with an alternative inducer of oxidative stress, diethyl-maleate (DEM), that depletes cellular glutathione, leading to accumulation of ROS (Sup. Fig. 5a, b); DEM has been used in numerous studies (including nuclear transport studies) to induce oxidative stress[71–75]. DEM was confirmed to increase ROS levels within ESCs by flow cytometric analysis (Sup. Fig. 5c). IPO13$^{+/+}$ and IPO13$^{-/-}$ ESCs were treated with 100–400 μM DEM for 24 h prior to flow cytometric analysis to assay for cell viability using PI (Sup. Fig. 5). DEM concentrations <200 μM did not induce cell death in either IPO13$^{+/+}$ or IPO13$^{-/-}$ ESCs, but higher concentrations (>300 μM) triggered the death of IPO13$^{+/+}$ and IPO13$^{-/-}$ cells, although the latter underwent significantly less death comparatively (c. 10 and 20% less death at 300 and 400 μM, respectively), confirming IPO13$^{-/-}$ were more resistant to cell death resulting from DEM-induced oxidative stress. Clearly, the contribution of IPO13 to oxidative stress-induced death is not limited to H$_2$O$_2$-induced oxidative stress, extending to the cellular oxidative stress response more generally.

**IPO13 interacts with and regulates SP1 and KLF4 nucleocytoplasmic distribution.** We next tested whether IPO13 could modulate nucleocytoplasmic transport of KLF4 and/or SP1. Due to very low transfection levels observed in the ESC lines, HeLa cells were used as a model to examine this in the absence or presence of siRNA-induced IPO13 knockdown and separately in the absence or presence of ectopically expressed IPO13. Well-characterised IPO13 import and export cargoes, UBC9[3] and EIF1A[76], shown to localise as previously (Baade et al. 2018)[77], were used as controls (Sup. Fig. 6). IPO13 knockdown resulted in a reduction/enhancement of nuclear localisation of UBC9 and EIF1A, respectively (Sup. Fig. 6a–d), as expected, while over-expression of IPO13 increased UBC9 nuclear accumulation and EIF1A cytoplasmic localisation (Sup. Fig. 6f–o), validating the system as a model. Subcellular localisation of ectopically expressed KLF4 was examined by CLSM in live HeLa cells pretreated with siRNA to knockdown IPO13 (Fig. 5a–c). KLF4 showed strong nuclear localisation (nuclear to cytoplasmic ratio, Fn/c of >100), which was significantly ($p < 0.0001$) decreased by H$_2$O$_2$-induced oxidative stress (Fn/c of c. 50), which was not observed in IPO13 siRNA treated cells. Importantly, the sublethal concentration of H$_2$O$_2$ used for imaging and interaction studies does not induce any significant levels of cell death in the HeLa cells (as examined by flow cytometry, Sup. Fig. 7a, b) and is confirmed to induce oxidative stress by increasing intracellular ROS levels (Sup. Fig. 7c). The clear implication is that IPO13 may act as a nuclear exporter for KLF4. To confirm this, we tested the effect of overexpression of DSRED-IPO13 on KLF4 localisation (Fig. 5d–e), which significantly diminished nuclear localisation in

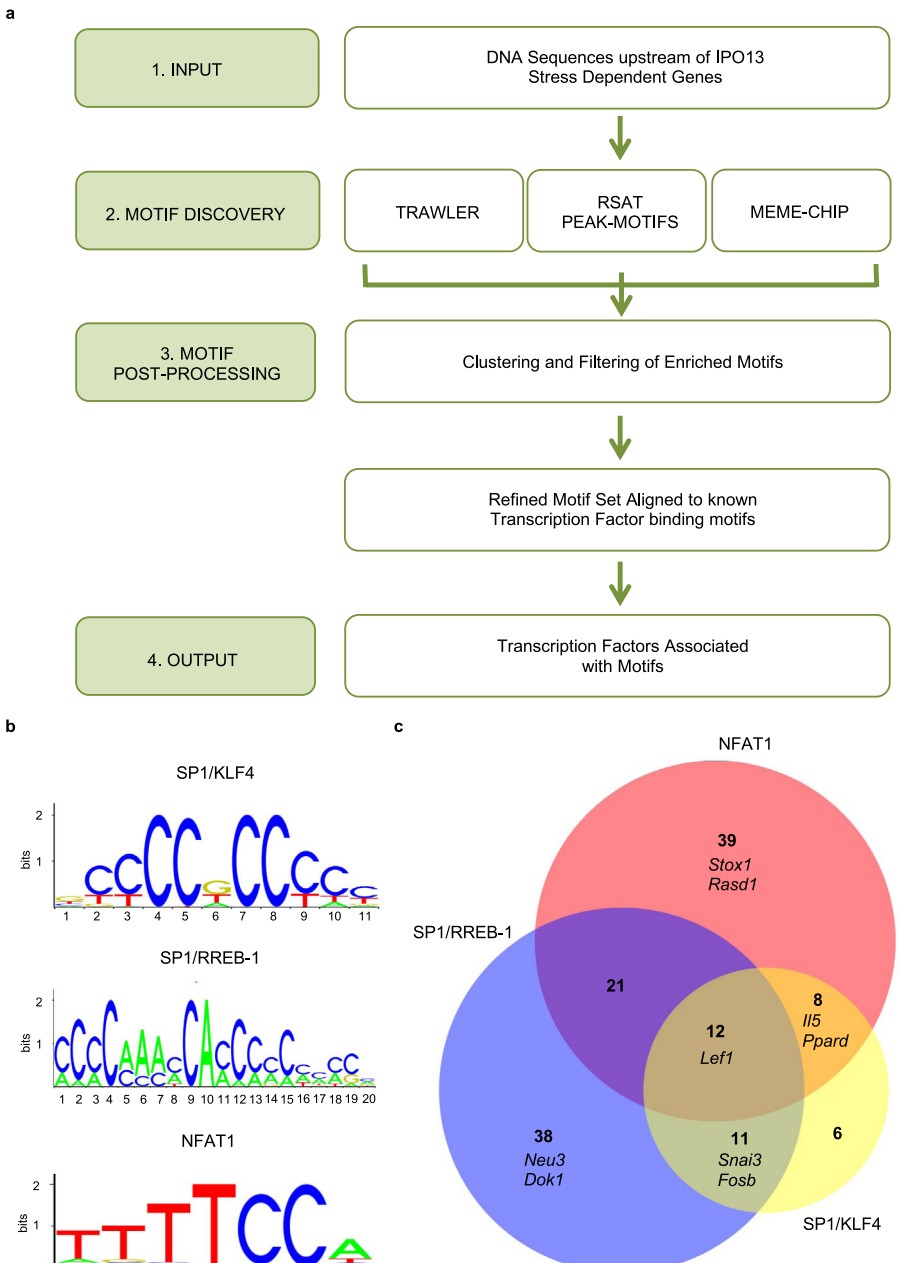

**Fig. 3 TF binding motif enrichment in the IPO13-dependent stress transcriptome contributing to the IPO13 dependent oxidative stress transcriptome.**
**a** Pipeline for motif discovery. Genome coordinates identified were derived from UCSC Mouse Encode from ChIP Seq experiments performed in mESCs for the H3K4me1, H3K4me3, and H3K27ac histone marks. Genomic regions enriched for these marks were associated with a gene using GREAT software from Stanford Bioinformatics, and then filtered to include only the 277 IPO13-dependent stress-regulated genes identified (see text). Overlapping genomic regions were merged using Galaxy before collecting the FASTA sequences for the final (1) set of genomic regions using UCSC Mouse Encode. FASTA sequences were input (2) into each of the Motif Discovery Tools Trawler, RSAT Peak-Motif, and MEME-CHIP to identify overrepresented motifs. Motifs identified by each tool were consolidated (3) and then inputted into STAMP to cluster and merge motifs, removing redundant or nonsense motifs. The final set of refined motifs were aligned to known TF binding motifs using the JASPAR library via STAMP. The output set (4) of TFs for each motif set was ranked by STAMP according to the similarity of the input motif to matched TF motif. **b** Candidate motifs are overrepresented in the IPO13 dependent stress transcriptome identified in (**a**) with consensus TFs binding them indicated. **c** Venn diagram of the genes containing one or multiple motifs from (**b**) in the IPO13 dependent stress transcriptome, with numbers of genes indicated in the different unique/overlapping parts of the diagram, together with the names of the genes selected for RT-qPCR validation i.e. *Stox1* and *Rasd1* contain the motif aligned to the NFAT1 binding motif, *Neu3*, and *Dok1* contain the motif aligned to the SP1 and RREB-1 binding motif, *Il5,* and *Ppard* contains the motif aligned to the NFAT1 binding motif and to the SP1 and RREB-1 binding motif, *Snai3,* and *Fosb* contain the motif aligned to the SP1 and RREB-1 binding motif and to the SP1 and KLF4 binding motif and *Lef1* contains the motif aligned to all three of the TF binding motifs. RT-qPCR validation for selected genes is shown in Sup. Figure 3.

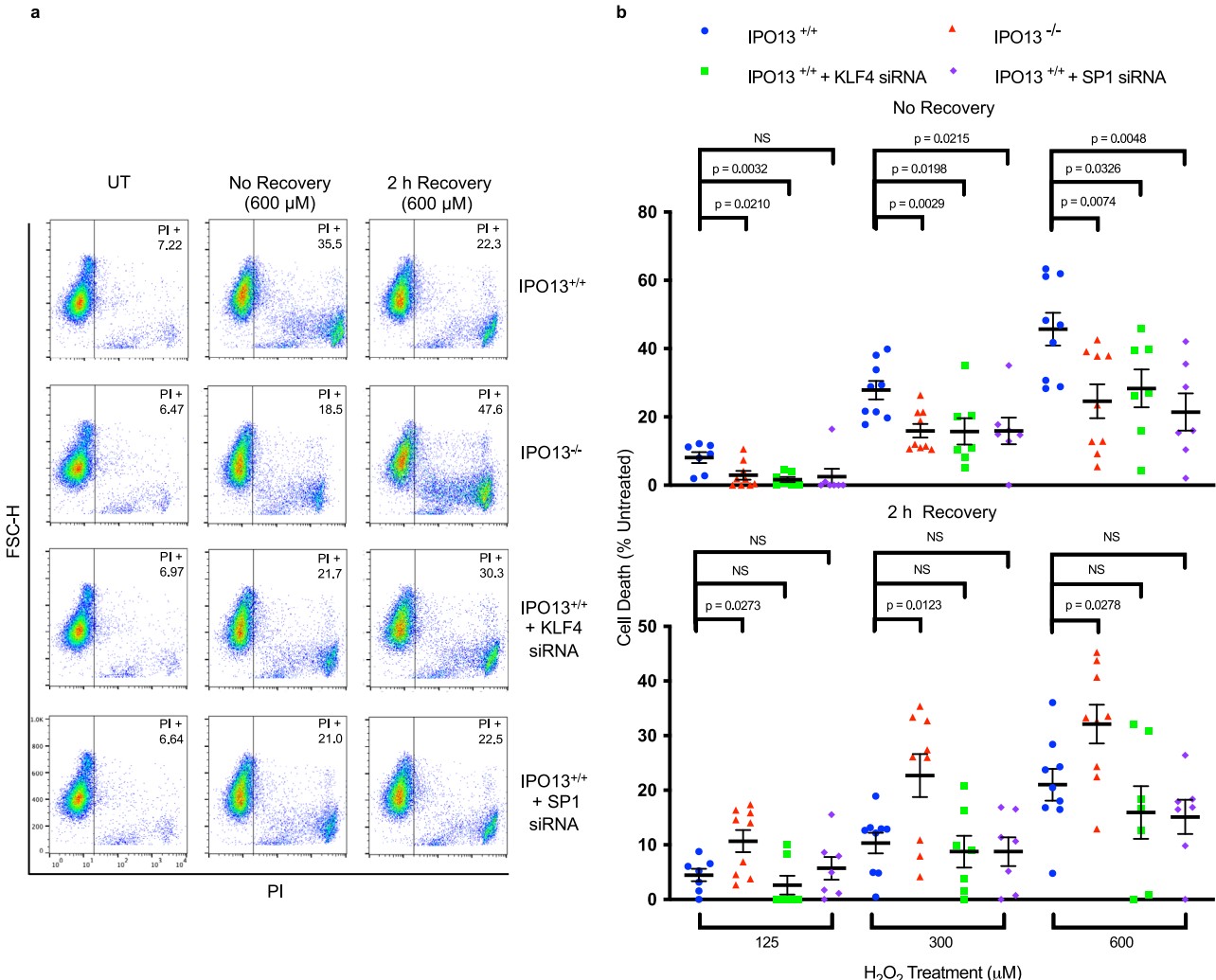

**Fig. 4 Kruppel-Like Factor 4 (KLF4) and Specificity protein-1 (SP1) contribute to IPO13 dependent oxidative stress induced cell death.** IPO13$^{+/+}$ ESCs were treated without or with siRNAs targeting SP1 or KLF4. 3 d post transfection, cells were treated with $H_2O_2$ as indicated for 1 h, and then stained either immediately ("No recovery") or subsequent to 2 h recovery as indicated, for propidium iodide (PI) and flow cytometric analysis. **a** Representative dot plots of the indicated samples with the percentage of ESC PI positive. **b** Pooled data (IPO13$^{+/+}$ 125 μM ($n = 7$), IPO13$^{+/+}$ 300–600 μM ($n = 9$), IPO13$^{-/-}$ 125–600 μM ($n = 9$), IPO13$^{+/+}$ +KLF4 siRNA 125–600 μM ($n = 7$), IPO13$^{+/+}$ + SP1 siRNA 125–600 μM ($n = 7$)) for % PI positive cells relative to untreated (UT) control from the analysis such as that in **a** are shown (mean ± SEM), where $p$ values (two tailed student's t test) top to bottom left to right are no recovery: $p = 0.6160$, $p = 0.0032$, $p = 0.0210$, $p = 0.0215$, $p = 0.0198$, $p = 0.0029$, $p = 0.0048$, $p = 0.0326$, $p = 0.0074$, and 2 h recovery: $p = 0.6160$, $p = 0.3882$, $p = 0.0273$, $p = 0.6300$, $p = 0.6484$, $p = 0.0123$, $p = 0.1919$, $p = 0.3598$, $p = 0.0278$. Source data are provided as a Source Data file.

either the absence or presence of $H_2O_2$ treatment (Fn/c of 37 and 19 respectively).

Subcellular localisation of SP1 was examined by fixation and immunostaining; endogenous SP1 (Fig. 5f–g) was strongly nuclear (Fn/c of c. 10) but this was decreased by $H_2O_2$ treatment (Fn/c of c. 4) or siRNA to IPO13 (Fn/c of c. 7). IPO13 knockdown also reduced the relocalisation of SP1 to the cytoplasm in response to $H_2O_2$ (Fn/c of c. 7 compared to a value of 4 in the control). The clear implication is that IPO13 may act as a nuclear exporter for SP1. To confirm this, we tested the effect of overexpression of GFP-IPO13 on SP1 localisation (Fig. 5h), with a clearly enhanced cytoplasmic accumulation of SP1 in cells whether subjected to $H_2O_2$-induced oxidative stress or not. Quantitative analysis (Fig. 5i) confirmed the results, with significantly ($p < 0.0001$) reduced nuclear accumulation (lower Fn/c) in the presence of overexpressed IPO13.

To verify that IPO13 is a nuclear exporter of KLF4 and SP1 beyond the HeLa cell line, we examined the subcellular localisation of endogenous KLF4 and SP1 in response to $H_2O_2$ induced oxidative stress in the IPO13$^{+/+}$ ESCs and how this compares in the IPO13$^{-/-}$ ESCs (Sup. Fig. 8a–e). In the IPO13$^{+/+}$ ESCs, endogenous KLF4 is distributed throughout the cell, however it is more nuclear than cytoplasmic (Fn/c of 1.1). After $H_2O_2$ treatment, KLF4 nuclear localisation is significantly decreased, becoming more cytoplasmic than nuclear (Fn/c of 0.6). The effect of $H_2O_2$-induced oxidative stress on KLF4 localisation is abrogated in the IPO13$^{-/-}$ ESCs. Endogenous SP1 was highly nuclear in the IPO13$^{+/+}$ ESC (Fn/c of 4.6) which was decreased in $H_2O_2$ treated cells (Fn/c of 2.7). In contrast, SP1 remained highly nuclear with or without $H_2O_2$ treatment in IPO13$^{-/-}$ ESC (Fn/c 4.6 and 4.3 respectively). Similarly, the export of ectopically expressed GFP-tagged KLF4 or GFP-tagged SP1 was abrogated in IPO13$^{-/-}$ compared to $H_2O_2$-treated IPO13$^{+/+}$ (Sup. Fig. 8f–i).

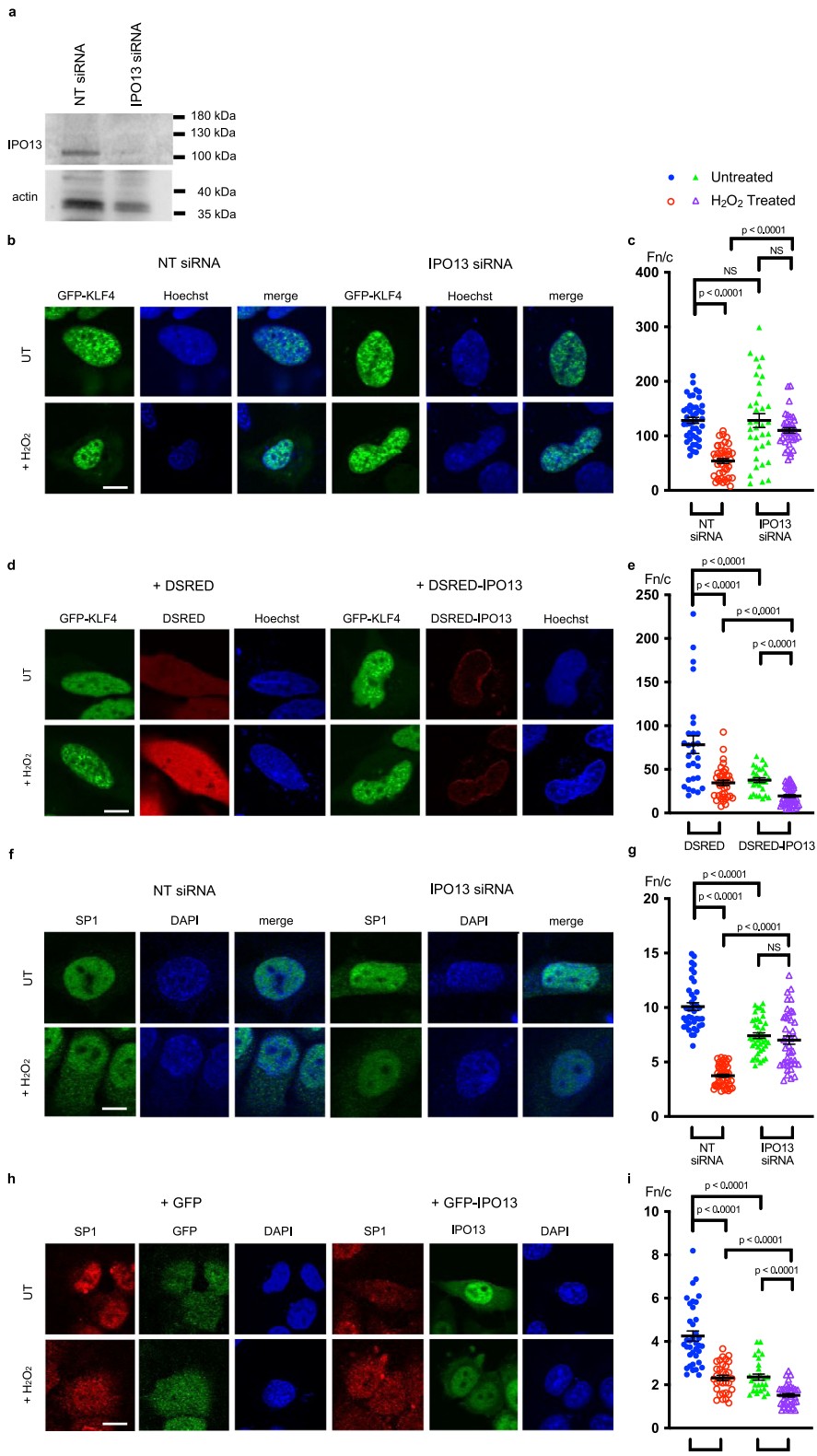

Together, the results suggest that IPO13 functions as a nuclear exporter for KLF4 and SP1 in both HeLa cell and ESC lines.

To establish that the nuclear export of KLF4 and SP1 by IPO13 is not limited to $H_2O_2$-induced oxidative stress, we re-examined the subcellular localisation of KLF4 and SP1 in HeLa treated with DEM (which was confirmed to induce oxidative stress by increasing intracellular ROS levels, see Sup. Fig. 7f) in HeLa cells with and without siRNA-induced IPO13 knockdown and HeLa cells with and without ectopic IPO13 expression (Sup. Fig. 9). Nuclear localisation of ectopic KLF4 was significantly reduced in response to DEM-induced oxidative stress (Sup. Fig. 9a–c). However, this effect was not seen in cells where IPO13 was

**Fig. 5 IPO13 can mediate nuclear import and export of KLF4 and SP1 respectively in response to H$_2$O$_2$- induced oxidative stress. a–c** HeLa cells were transfected with non-targeting (NT) or IPO13 targeting siRNA 72 h prior to treatment with 125 μM H$_2$O$_2$ for 1 h. Cells were transfected to express GFP-KLF4 and processed for CLSM imaging and/or Western analysis. Cells were stained with Hoechst to highlight nuclei. **a** Total cell extracts were subjected to Western analysis using rabbit-anti-IPO13 (Protein Tech), with mouse-anti-actin (Abcam) as a control, and imaged using the ChemiDoc Gel Imaging System (Biorad). **b** Representative images of cells transfected to express GFP-KLF4. Scale bar = 10 μm. **c** Quantitative analysis of ectopic KLF4 protein localisation was carried out using the ImageJ software on images such as those in (**a**) to determine the nuclear to cytoplasmic ratio (Fn/c). Data represent the mean ± SEM for 44 cells (NT Untreated (UT)), 41 cells (NT+H$_2$O$_2$ Treated), 35 cells (IPO13 siRNA UT), and 37 cells (IPO13 siRNA + H$_2$O$_2$ treated) respectively for the nuclear (Fn) and cytoplasmic fluorescence (Fc) above background fluorescence. *p* values (two-tailed student's *t*-test) top to bottom: *p* < 0.0001, *p* = 0.1789, *p* = 0.9899, and *p* < 0.0001. **d–e** HeLa cells were co-transfected to express either DSRED or DSRED-tagged IPO13 with GFP-KLF4 and treated for 1 h ± 125 μM H$_2$O$_2$ prior to CLSM imaging. Cells were stained with Hoechst to highlight nuclei. **d** Representative images of cells co-expressing DSRED or DSRED-IPO13 with GFP-KLF4. Scale bar = 10 μm **e**. Quantitative analysis of ectopic KLF4 protein localisation performed as per (**c**). Data represent the mean ± SEM for 28 cells (DSRED UT), 38 cells (DSRED + H$_2$O$_2$ Treated), 27 cells (DSRED-IPO13 + UT) and 44 cells (DSRED-IPO13 + H$_2$O$_2$ Treated). p values (two-tailed student's *t*-test) top to bottom: *p* < 0.0001, *p* < 0.0001, *p* < 0.0001, and *p* < 0.0001. **f–g** HeLa cells were transfected with siRNA as in **a–c**. prior to treatment with 125 μM H$_2$O$_2$ for 1 h. Cells immediately processed for CLSM imaging and immunostained with anti-SP1. **f** Representative images of cells immunostained with anti-SP1. Scale bar = 10 μm **g**. Quantitative analysis of endogenous SP1 localisation carried out as per (**c**). Data represent the mean ± SEM for 39 cells (NT UT), 47 cells (NT + H$_2$O$_2$ Treated), 41 cells (IPO13 siRNA UT), and 43 cells (IPO13 siRNA + H$_2$O$_2$ Treated). p values (two-tailed student's t-test) top to bottom: *p* < 0.0001, *p* < 0.0001, *p* < 0.0001, and *p* = 0.2018. **h–i** HeLa cells were co-transfected to express either GFP or GFP-tagged IPO13 and treated for 1 h ± 125 μM H$_2$O$_2$ prior to immunostaining using anti-SP1 antibody. **h** Representative images of cells immunostained with anti-SP1. Scale bar = 10 μm. **i** Quantitative analysis of endogenous SP1 localisation carried out as in (**c**). Data represent the mean ± SEM for 37 cells (GFP UT), 32 cells (GFP + H$_2$O$_2$ Treated), 29 cells (GFP-IPO13 UT), and 38 cells (GFP-IPO13 + H$_2$O$_2$ Treated). *p* values (two-tailed student's *t*-test) top to bottom: *p* < 0.0001, *p* < 0.0001, *p* < 0.0001 and *p* < 0.0001. Source data are provided as a Source Data file.

knocked down. Further, overexpression of IPO13 significantly reduced KLF4 nuclear localisation in both the absence of DEM, and to an even greater extent in the presence of DEM (Fn/c of 40 and 20 respectively) (Sup. Fig. 9d–f). This data suggests that KLF4 is exported from the nucleus in response to DEM-induced oxidative stress and that IPO13 is the nuclear exporter that enables this export.

The subcellular localisation of SP1 was also effected by DEM treatment whereby DEM-induced nuclear export of endogenous SP1 (Fn/c of 6–3) (Sup. Fig. 9g–i). However, export of SP1 from the nucleus in response to DEM treatment was not observed in the siRNA-induced IPO13 knockdown cells (Fn/c of 4), suggesting that IPO13 also exports SP1 from the nucleus in response to DEM-induced oxidative stress. To confirm this, endogenous SP1 localisation was examined in HeLa with ectopic IPO13 expression in the presence and absence of DEM. Significantly, ectopic expression of IPO13 resulted in a greater still DEM-induced accumulation of SP1 in the cytoplasm compared to where IPO13 was not overexpressed (Sup. Fig. 9j, k). Together, this data indicates that IPO13 functions as a nuclear transporter for KLF4 and SP1 and is not limited to H$_2$O$_2$-induced oxidative stress and thus suggests that it performs as a critical regulator of nuclear transport during oxidative stress.

To confirm the interaction of IPO13 with KLF4 or SP1 in a cell context, GFP-tagged KLF4 (Fig. 6a–c) or SP1 (Fig. 6d–f) were immunoprecipitated from cells ectopically expressing DSRED-IPO13 in the absence or presence of H$_2$O$_2$ treatment; the dependence on Ran of IPO13 binding to KLF4 or SP1 was addressed in parallel by performing immunoprecipitations in the absence or presence of GTPγS (a non-hydrolysable form of GTP that maintains Ran in the GTP-bound state[78]). IPO13 co-precipitated with both GFP-KLF4 and GFP-SP1 with or without H$_2$O$_2$-induced oxidative stress, but not with GFP alone, indicating the specificity of the interactions. Notably, IPO13 co-precipitated with GFP-KLF4 and GFP-SP1 to a greater extent in H$_2$O$_2$ treated cells (Fig. 6b, e). Further, the addition of GTPγS resulted in an approximately 2-fold increase in co-precipitation of IPO13 with GFP-KLF4 (Fig. 6b, c) and GFP-SP1 (Fig. 6e, f) in H$_2$O$_2$ treated cells, implying RanGTP may facilitate IPO13 binding in both cases. Importantly, the data overall demonstrate that IPO13 both interacts with and facilitates the nuclear export of KLF4 and SP1.

## Discussion

This is the first study to delineate a specific role for IPO13 in response to stress, which appears to be at least in part related to its nuclear export role with respect to TFs such as SP1 and KLF4, not previously known to be IPO13 cargoes, that are themselves key players in stress responses through nuclear action as transcriptional activators or repressors with respect to target genes to effect up- or down-regulation to impact cell survival/trigger further transcriptomic response. Through mediating nuclear transport of TFs such as SP1 and KLF4 under conditions of stress, IPO13 enhances cell susceptibility to oxidative stress. Stress-activated signalling molecules that potentiate the cellular response to stresses (eg. oxidative stress) need to be in "the right place at the right time" in order to trigger downstream stress response pathways appropriately and efficiently. Excitingly, this study establishes a key mechanism by which proteins involved in stress response can travel between the nuclear and cytoplasmic compartments of the cell, and begins to unravel the details of IPO13's potential to regulate multiple stress response pathways impacting cell survival.

Importantly, our results challenge the paradigm that cellular stress, inclusive of oxidative stress, results in large-scale abrogation of IPO-dependent nuclear transport[21–25,79]. This has largely been attributed to the dependency on the Ran protein gradient for efficient recycling of IPOs from the nucleus to the cytoplasm for future rounds of nuclear import[80], and the fact that cellular stress appears to perturb the Ran gradient[81,82], redistributing Ran to the cytoplasm and in turn resulting in a mislocalisation of nuclear transport factors[21]. We show that IPO13 is different, in that it is able to traffic specific cargoes out of the nucleus under oxidative stress conditions. IPO13's ability to continue to mediate nuclear transport even when the RanGTP gradient has collapsed presumably relates to its ability to function as a transporter in the absence of RanGTP[83]. Size exclusion chromatography has been used, for example, to show that export cargo EIF1A can bind to IPO13 even in the absence of RanGTP[76]. Further, although RanGTP can clearly trigger import cargo dissociation from IPO13 in vitro (and based on the results here in cell lysates for KLF4 and SP1 — see Fig. 6 — potentially enhance nuclear export cargo binding), certain import cargoes appear to require binding of an export cargo for complete dissociation[83]. This competitive displacement mechanism of nuclear import cargo by binding of

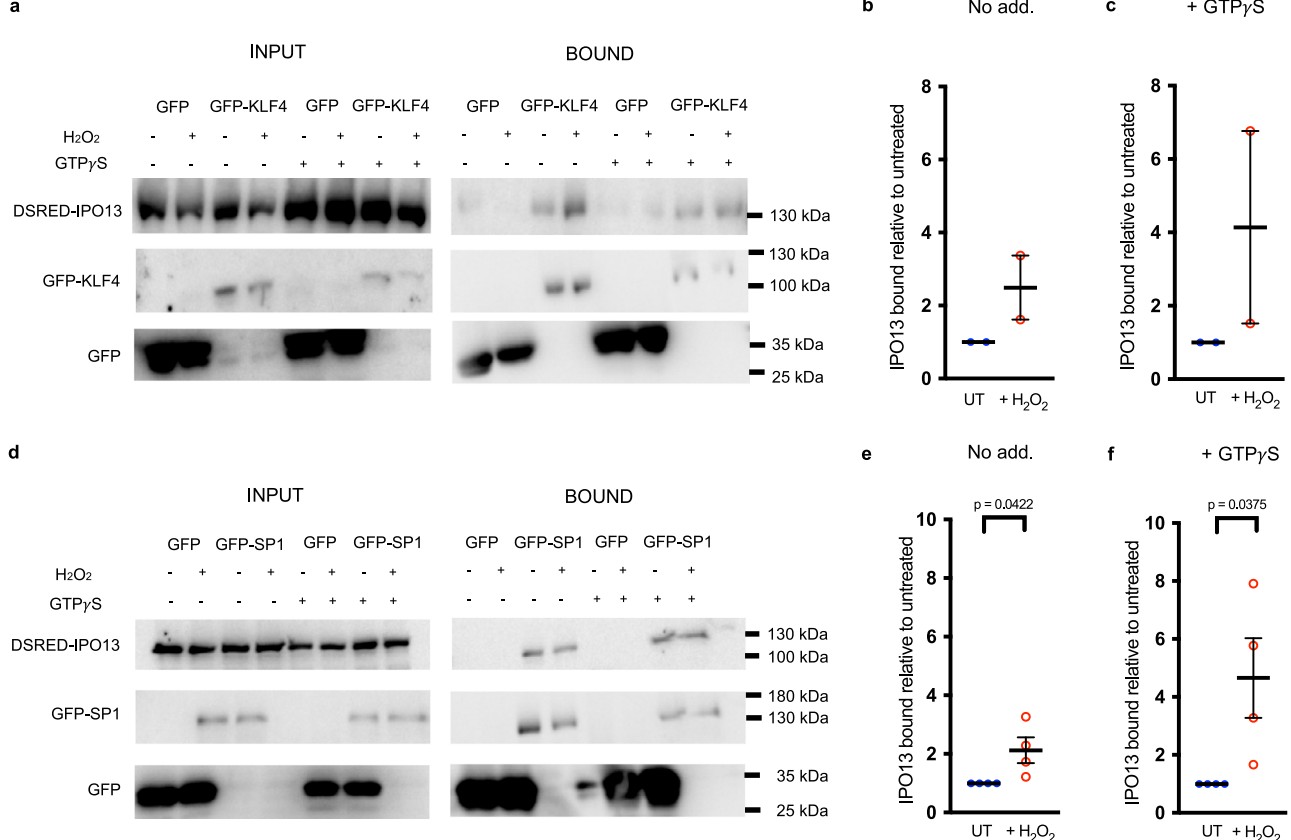

**Fig. 6 KLF4 and SP1 interact with IPO13 in the presence and absence of $H_2O_2$-induced oxidative stress.** HeLa cells transfected to co-express GFP or GFP-KLF4 (**a**–**c**) or -SP1 (**d**–**f**) with DSRED-IPO13. At 16 h post-transfection, cells were treated with or without 125 μM $H_2O_2$ for 1 h before lysis and incubation with or without GTPγS and immunoprecipitation (IP) using GFP-Trap (Chromotek). Input/IP samples were probed by Western blotting using mouse-anti-GFP (Roche) or rabbit anti-IMP13 (Protein Tech) antibodies and then imaged using the ChemiDoc Gel Imaging System (Biorad). **b**, **c**, **e**, **f** Densitometric analysis was performed on images such as those in **a** and **d** for binding of DSRED-IPO13 to GFP-KLF4 (**a**) or GFP-SP1 (**d**) under $H_2O_2$ treated conditions and untreated (UT) conditions with and without the addition of GTPγS. The amount of co-immunoprecipitated IPO13 was normalized to the amount of protein available for co-immunoprecipitation (as seen in the input panels). This value was normalized to the amount of immunoprecipitated GFP-KLF4 or –SP1. Pooled results representing the mean ± SEM (error bars) for IPO13 bound under $H_2O_2$ treated conditions relative to UT conditions (no GTPγS treatment - No Add.) (**b** n = 2 independent experiments, **e** n = 4 independent experiments) or with GTPγS treatment under UT and treated conditions (**c** n = 2 independent experiments, **f** n = 4 independent experiments) are shown. p values (two-tailed student's t test) left to right: p = 0.0422 and p = 0.0375. Source data are provided as a Source Data file.

nuclear export cargo and vice versa suggests that IPO13's requirement for RanGTP to function in nuclear import/export roles is not absolute. These properties of IPO13 make it a unique member of the IPO family, able to transport protein cargo under stress conditions. Our data demonstrating that IPO13 is not markedly mislocalised under oxidative stress, but retains nuclear and cytoplasmic distribution, is consistent with this idea of reduced dependence on Ran for IPO13 function.

In summary, our use of an established ESC knockout model[4] and Next Generation Sequencing has made it possible to demonstrate that IPO13, additional to its acknowledged roles in development, is a key factor in the response to oxidative stress, being critical to transcriptional networks including those that regulate for cell death. Through being key cargoes of IPO13's nuclear export function that can remain functional during stress, the SP1 and KLF4 TFs are central to the IPO13-dependent oxidative stress response, which is likely to be critically important during early tissue development. The precise mechanism by which IPO13-dependent responses to stress contribute to cell repair/cell death in a developmental context is a focus of future work in this laboratory.

## Methods

**Cell culture, transfection, and immunofluorescence.** The IPO13[+/+] and IPO13[−/−] mESC lines have been described previously in detail[4]; they were maintained in feeder-free culture conditions in DMEM supplemented with 12.5% FBS, 1× Glutamax, 0.1 mM NEAA, 0.1 mM β mercaptoethanol, and 1000 U/ml LIF at 37 °C on 0.1% gelatin-coated surfaces in 5% $CO_2$ atmosphere in a humidified incubator Cells were passaged every 2 d. Cells of the HeLa human cervical epithelial cell line (ATCC® CCL-2™) were maintained in Eagle's Minimum Essential Medium (EMEM) containing 10% FBS, 1X Glutamax, 1× Nonessential Amino Acids (NEAA) at 37 °C in a 5% $CO_2$ atmosphere in a humidified incubator. Plasmid transfection of HeLa cells was performed using FuGene® HD Transfection Reagent (Promega) according to the manufacturer's instructions. Plasmid transfection of mESCs was performed using LIPO2000 Transfection Reagent (ThermoFisher) as per the manufacturer's instructions. The Dharmacon ON-TARGETplus siRNA system (GE Life Sciences) with DharmaFECT 1 transfection reagent was used for HeLa and ESCs as per the manufacturer's instructions. Predesigned siRNAs targeting KLF4 (SMARTpool L-040001-01-0005), SP1 (SMARTpool L-040633-02-0005), NFAT1 (SMARTpool L-054724-01-0005), RREB-1 (SMARTpool L-045323-01-0005), and IPO13 (SMARTpool L-020212-01) were used, with a non-targeting (SCRAM siRNA) control pool (D001810-10-50) as the negative control. For microscopy, HeLa cells (and mESCs) were seeded onto glass coverslips one day prior to imaging. $H_2O_2$ (Sigma) or DEM (ACROS Organics) was diluted freshly in growth medium immediately before use; cells were treated for 1 or 24 h respectively at the indicated concentration and were immunostained as previously described[78]. Briefly, cells were fixed with 4% paraformaldehyde (Sigma) after transfection +/− $H_2O_2$ or DEM treatment, and

permeabilised with 0.2% Triton X-100 (Sigma). Non-specific binding sites were blocked with blocking buffer (1% BSA in PBS) and immunostained with primary antibodies at room temperature for 1 h followed by 1 h incubation with conjugated secondary antibodies. Coverslips were mounted to glass slides with ProLong Gold antifade reagent with DAPI (Invitrogen).

**Antibodies**. Rabbit polyclonal anti-SP1 (Cat no. ABE135 Millipore, Western Blot (WB); 1:2000 and Immunofluorescence (IF); 1:200), anti-IPO13 (Cat no. 11696-2-AP, Protein Tech, WB; 1:1000), anti-KLF4 (Cat no. ab75486, WB; 1:2000, IF; 1:100 and ab129473, Abcam, WB; 1:500, IF; 1:100) and anti-EIF1A (Cat no. 38976, Abcam, 1:200) antibodies, and mouse monoclonal anti-NFAT1 (Cat no. ab2722 1:1000, Abcam), anti-Actin (Cat no. ab3280, Abcam, 1:4000), anti-UBC9 (Cat no. 610749, BD, 1:100) and anti-GFP (Cat no. 11814460001, Roche, 1:5000) antibodies were used as indicated. Alexa-Fluor 568-goat anti-rabbit (1:1000), Alexa-Fluor 568-goat anti-mouse (1:1000), Alexa-Fluor 488-goat anti rabbit (1:1000) and Alexa-Fluor 488-goat anti mouse (1:1000) IgG secondary antibodies were from Invitrogen.

**Plasmid constructs**. Plasmid pDsRed2-fused IPO13 for mammalian cell expression of IPO13 fused to the DsRed2 fluorescent protein was constructed by digestion of plasmid vector pDsRed2-C1 (Clontech, Mountain View, CA, USA), and insertion of polymerase chain reaction (PCR) generated IPO13 insert at EcoRI and SmaI restriction sites. Plasmid for mammalian expression of GFP-fused IPO13 was generated as described previously[4]. Plasmid for mammalian cell expression of the fusion protein GFP-KLF4 was constructed by inserting PCR amplified sequences encoding full-length mouse KLF4 in-frame into the BamHI and HindIII restriction sites of plasmid pEGFP-C1 C-terminal to the coding sequence of eGFP. The plasmid for mammalian cell expression of GFP-SP1 was from Jan-Jong Hung (National Cheng Kung University, Taiwan). The integrity of constructs was verified by DNA Sequencing.

**RNA extraction and sequencing**. Total RNA was isolated from cell lines using Isolate II RNA Mini Kit (Bioline) according to the manufacturer's instructions and RNA concentration was measured using the Qubit Fluorometric Quantitation (Invitrogen). Libraries were constructed using the Illumina TruSeq mRNA Preparation Kit with 1 μg of input RNA and the libraries were sequenced in three lanes of an Illumina HiSeq2,000 using paired-end 101 bp sequencing configurations. Sequenced reads were mapped to mouse genome reference mm10. Across all samples and replicates, c. 84% of the >80 million reads could be mapped to exons. Analysis of differential gene expression amongst ESC samples was undertaken using Voom/Limma as previously described[84] via Degust (v. 0.20), where the euclidean distances between all sample pairs were calculated from the log-cpm of the 200 most variable genes and approximated in a Multi-Dimensional Scaling (MDS) plot for Principle Component Analysis. The multi-dimensional scaling plot in Fig. 1b indicates the high reproducibility of the transcriptome profiling for the duplicates analysed for each condition. Genes were considered significantly differentially expressed where the FDR <0.05 and the change in expression was 2-fold or higher. Output from Degust was used to create the Venn diagrams that emphasise the differentially expressed genes that are unique or shared amongst two sample conditions. Pathway enrichment analysis was performed using the Functional Annotation Tool of the Database for Annotation Visualization and Integrated Discovery (DAVID) Bioinformatics Database (Version 6.8).

**Real-time qPCR**. Total RNA for RT-qPCR was isolated from cell lines using Isolate II RNA Mini Kit (Bioline) according to the manufacturer's instructions. RNA was reverse-transcribed by first-strand cDNA synthesis using the Superscript III Reverse Transcriptase/random hexamers kit (Invitrogen). cDNA was then subjected to qRT-PCR using the SensiMix SYBR Master Mix (Bioline Reagents, London, UK); primer sequences are listed in Sup. Table 1. Quantitative RT-PCR was performed using the C1000 Touch Thermal Cycler (Biorad) for triplicate reactions to detect and compare expression levels between IPO13$^{+/+}$ (with and without KLF4, SP1, RREB-1, or NFAT1 knockdown) or IPO13$^{-/-}$ ESC with and without 1 h $H_2O_2$ treatment, followed by 2 h recovery. Biorad CFX Manager 3.1 was used to analyse the data. Data were normalised to the expression of the housekeeping genes *Tbp* or *Sdha* using the ΔΔCt method as described[85].

**Motif enrichment analysis**. Motif discovery tools used in this study were Trawler[44,45], RSAT Peak-Motif[12,43] and MEME-ChIP[41]. The input data set for the tools is a FASTA formatted file of promoter and enhancer DNA sequence regions associated with genes within our 277-gene subset. To generate this input file, we collected genome coordinates from ChIP Seq experiments performed in mouse embryonic stem cells for histone marks H3K4me1 ('GSM769009'), H3K4me3 ('GSM769008'), and H3K27ac ('GSM1000099'). The peaks for these histone marks were collected from UCSC Mouse Encode[40] using the Track Search tool. Genome coordinates were annotated to a gene using GREAT software as described previously[86], and filtered to include only coordinates associated with our gene subset of interest. Overlapping genome regions in the subset were merged using the Merge tool in the Galaxy platform as described[87]. Merged genomic regions were utilised to collect FASTA sequences of the genome regions of interest using the Table Browser in UCSC Mouse Encode. The output of this are FASTA formatted files of the DNA

sequences upstream of genes within our 277 IPO13 Dependent Stress Gene subset.

FASTA sequences were input into the motif discovery tools above, and motifs identified were consolidated and converted to formats recognised by the DNA motif comparison tool STAMP[46] using the default parameters. Motifs were clustered into families based on alignment, and the families of motifs identified by multiple discovery tools run through STAMP separately and aligned to known TF binding motifs by STAMP using the JASPAR v2010 database. TF binding motifs were aligned to the output list of motifs, and ranked according to the similarity of consensus and input motif similarity.

**Flow cytometry**. Cell viability was assessed using propidium iodide (PI) staining/flow cytometry. ESCs and HeLa were seeded in 6-well plates and cultured overnight in experiments without transfection as well as with GFP-tagged protein transfection or for 3 d with siRNA targeting the indicated gene, prior to treatment with $H_2O_2$ for 1 h or DEM for 24 h without or with recovery in fresh growth medium lacking $H_2O_2$ or DEM. ROS were detected in ESCs and HeLa using CellROX Green Reagent (ThermoFisher) according to the manufacturer's instructions. Briefly, cells were seeded on a 6-well plate. After $H_2O_2$ or DEM treatment, CellROX Green was added to the culture media for 30 mins at a final concentration of 5 or 10 μM and incubated at 37 °C in the dark. In either case, cells were harvested for flow cytometry by trypsinisation/centrifugation, resuspended in 0.5 ml ice-cold PBS supplemented with 12.5% FBS, and stained with PI (Invitrogen) at a final concentration of 1 μg/ml where appropriate. PI-positive cells (minimum of 15,000 cells/sample) or CellROX Green stained cells were quantified by flow cytometry (FACS Calibur using CELLQuest Version 3.3), and analysed using the FlowJo software (Version 10.7.2). The gating strategy used is shown in Sup. Fig. 10.

**Confocal laser scanning microscopy (CLSM) and image analysis**. Fixed cells immunostained for endogenous proteins or live cells ectopically expressing fusion proteins were imaged using a 60× oil immersion objective lens on a FluoView FV1000 Confocal Microscope. The nuclear to cytoplasmic fluorescence ratio (Fn/c) was determined as previously[88] using the equation Fn/c = (Fn−Fb)/(Fc−Fb), where Fn is the nuclear fluorescence, Fc is the cytoplasmic fluorescence and Fb is the background due to autofluorescence from digitized images using the Image J 1.48 v public domain software (NIH).

**Coimmunoprecipitation (Co-IP) and western blotting**. HeLa cells expressing GFP fusion proteins together with dsRed2-IPO13 treated with and without $H_2O_2$ were lysed 12 h post-transfection. In some samples, lysates were preincubated with GTPγS at a final concentration of 1.7 mM (Sigma-Aldrich) for 20 min on ice. Immunocomplexes were precipitated and eluted using GFP-Trap resin (Chromo-Tek) according to manufacturer instructions. Input and immunoprecipitations were analysed by Western blotting.

**Statistical analysis**. Statistical analysis was performed using the 2-tailed Student's *t*-test, assuming equal variances between two groups (Prism 8 Software), unless otherwise stated. A confidence level of 95% ($p < 0.05$) was considered a statistically significant difference.

**Reporting summary**. Further information on research design is available in the Nature Research Reporting Summary linked to this article.

## Data availability

The RNA sequencing data generated in this study has been submitted to the Gene Expression Omnibus (GEO) database under accession number ("GSE108913"). Database and tools used in this study are publicly available; "RSAT Peak-Motif [http://rsat.sb-roscoff.fr/peak-motifs_form.cgi]", "Trawler [https://trawler.erc.monash.edu.au]", "MEME-ChIP [https://meme-suite.org/meme/tools/meme-chip]", "STAMP [http://www.benoslab.pitt.edu/stamp/]", "JASPAR [http://jaspar.genereg.net]", "GREAT [http://great.stanford.edu/public/html/index.php]", "Galaxy [https://usegalaxy.org]", "DAVID [https://david.ncifcrf.gov]" and "UCSC Mouse Encode (Track Search, Table Browser) [https://genome.ucsc.edu/ENCODE/]". All other relevant data supporting the key findings of this study are available within the article and its Supplementary Information files or from the corresponding author upon reasonable request. Source data are provided with this paper.

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

## Acknowledgements

We thank Stuart Archer from the Monash Bioinformatics Platform for advice on RNA-seq, Kim Lieu for the pDsRed2-IPO13 plasmid, Monash Micro Imaging Facility, and Monash Micromon Sequencing Facility for assistance with microscopic imaging and sequencing, respectively. We acknowledge the financial support of the National Health and Medical Research Council Australia (fellowship APP1002486/APP1103050), the National Breast Cancer Foundation (fellowship CDA-17-007), National Health and Medical Research Council/Heart Foundation Career Development Fellowship (1049980), and MNHS Platform Access Grant (PAG15-0016). K. A. G. acknowledges the scholarship support from Monash University (MBio Discovery Scholarship). The Australian Regenerative Medicine Institute is supported by grants from the State Government of Victoria and the Australian Government. This research was supported by the use of the Nectar Research Cloud, a collaborative Australian research platform supported by the National Collaborative Research Infrastructure Strategy (NCRIS).

## Author contributions

K.A.G., K.M.W., and D.A.J. designed the studies, analysed the experiments, and wrote the paper. M.R. provided expert knowledge and guidance. K.A.G., D.A.J., and M.R. performed computational analysis of the sequencing data. K. A.G. and H.L. conducted qRT-PCR analysis, and K.A.G. conducted and completed all other experiments and analyses. All authors reviewed the results and approved the paper.

## Competing interests

The authors declare no competing interests.
