## [Peer Review File · Nature Communications]

Reviewers' comments:

Reviewer #1 (Remarks to the Author):

The authors investigated genes regulated by IPO13 under short exposure of lethal concentrations of hydrogen peroxide. They performed RNA seq. by comparing IPO13 WT with KO mESC. Further promoter motif analyses of IPO13-regulated genes suggested the enrichment of genes regulated by SP1 and KLF4. IPO13 KO cells as well as WT mESC with SP1 or KLF4 knockdown were more resistant to high concentrations (125-600 uM) of hydrogen peroxide. IPO13 interacts with KLF4 and SP1 and may work as a nuclear importer of KLF4 and an exporter of SP1.

Major

1. The authors tend to interpret the acute toxicity of extremely high concentrations of hydrogen peroxide as oxidative stress. The physiological relevance is questionable.
2. As the treatment of hydrogen peroxide is called oxidative stress (or sometimes just called stress) throughout the manuscript including the title, other conditions triggering production of ROS and/or causing oxidative stress should be tested in Figures 4 and 5 along with providing experimental evidence of oxidative stress.
3. Since knockdown of SP1 or KLF4 is not equivalent or mimicking the deficiency of IPO13, the approach in Figure 4 and author's interpretation that SP1 and KLF4 both contribute to IPO13-dependent oxidative stress induced cell death (lines 222-223) sound incorrect or misleading. Thus, the molecular mechanism through which the deficiency of IPO13 makes mESC resistant to the toxicity of hydrogen peroxide is not clear.
4. Along the same lines, (if the authors would not agree with the point 3), the reproducibility of Figure 4 should be assessed in other cell types.
5. Rescue experiments are also necessary by introduction of IPO13 and IPO13-regulated genes back to IPO13 KO cells and assessment of susceptibility to hydrogen peroxide.
6. IPO13 interacting proteins were previously characterized by Y2H screening (Fatima et al., 2017, Ref.#4), in which SP1 or KLF4 was not found as an IPO13 binding protein. So, these are new findings. The approach in this work is also different from the previous work; however, due to the lack of connection between SP1 or KLF4 target genes and the mechanism of susceptibility to toxicity of hydrogen peroxide in conjunction with IPO13, the overall of this work is more descriptive than mechanistic; therefore its contribution to the field may be incremental at most.

Minor

1. Switching cells from mESC to HeLa cells in Fig. 5 is not well justified.
2. Line 63-: this paragraph is pretty much repetitive of the abstract. The introduction should cover more background and roles of IPO in (oxidative) stress response tied with the rationale behind this work.
3. Line 123 about CYP2S1, downregulated in response to what stress? References should be cited there.
4. Line 170 - apoptosis regulation should be clarified as pro- or anti-apoptotic.
5. All Western blots need MW markers.

Reviewer #2 (Remarks to the Author):

This manuscript describes roles of importin 13 in the oxidative stress response through its nuclear

transport activity. The transcriptomic analysis of wild-type and importin 13 knockout mouse embryonic stem cells (mESCs) indicates enrichment of differentially expressed genes in stress responses and regulation of apoptosis. Analysis of importin 13-dependent genes responsive to oxidative stress show proteins enriched with motifs aligned to consensus sites for the transcription factors specificity proteins 1 (SP1) or Kruppel like factor 4 (KLF4). Stress sensitivity differ among importin 13 knockout cells and wild type cells in cells knocked down with SP1 or KL4. The authors show effects of nuclear import or nuclear export of SP1 and KLF4 in importin 13 knocked down cells or importin 13 over expressed cells. They also show binding of importin 13 with SP1 or KLF4.

Identification of cellular role(s) of specific nuclear transport pathway is very important, however such study is still rare, due to the complexity of substrates that bind to each nuclear carrier proteins. In this sense, present study that demonstrate cellular function of importin 13 is very important and valuable. The advantage of this study is 1) careful transcriptomic analysis and 2) attempt to link transport substrate of importing 13.

Although overall study of manuscript is excellent, I have two concerns as below.

- 1) The authors claim importin 13 mediates nuclear import of KLF4 and nuclear export of SP1, but images in Figure 5 are unclear (especially, Fig5a, f + IPO3 untreated). Merging images might help. In addition, already established nuclear import substrate(s) and nuclear export substrate(s) should be shown as a positive control. Also, expression levels of proteins (SP1, KLF4) and over expressed levels of importin 13 should be examined by western blotting.
- 2) Ran GTP dependency of KLF4 and SP1 binding to importin 13 must be examined.

Reviewer #3 (Remarks to the Author):

In this manuscript the authors demonstrate that IPO13 plays a specific role in stress dependent nuclear transport of key transcription factors, including KLF4 and SP1. The authors start with molecular analysis of identification of IPO13-dependent oxidative stress transcriptome and regulated transcriptional network in mESC, including evaluation of the contribution of two key transcription factors, KLF4 and SP1 in IPO13-dependent cell death, then switch to IPO13-mediated nuclear import and export of KLF and SP1, respectively in HeLa cells. Overall, the authors provide novel data and sufficient evidence that IPO13 is a key factor in mediating a response to oxidative stress via nuclear import/export of critical transcription factors, KLF4 and SP1, that regulate cell death.

Although the provided data could be of interest to others in the field and the scientific community in general several concerns need to be addressed before publication in Nature Communications could be granted.

A main concern of this reviewer is that the authors do not provide a rationale for switching from mESC to HeLa cells for the IPO13-dependent import/export and interaction studies presented in Figures 5 and 6? What impact on the authors' data interpretation and conclusions would a potential difference in the mechanistic and gene-specific role(s) of IPO13 between mES and HeLa cells have? Differences in IPOs and their function between cells and tissues including ESCs and differentiated progenies, clearly exists (see for example the review by Belmonte and colleagues (Yang et al. Trends Mol. Med. 2014, 20(1): 1-7)

Another concern lies with regard to Suppl Fig 3: This reviewer's understanding of the figure legend "IPO13+/+ ESCs were treated without or with siRNA targeting [gene name]" is that untreated IPO13+/+ ESCs were compared to siRNA treated ESCs. This understanding seems underscored by the figure itself, which says "IPO13+/+ and not SCRAM siRNA vs siRNA [gene name]". This is not an appropriate comparison. Instead IPO13+/+ cells treated with the non-targeting (SCRAM siRNA)

control pool (D001810-10-50) are the appropriate control, as the authors have done for Fig. 5.

Additional comments:

Fig 5f and g are referred to before 5d and e, thus the order should be switched in the figure.

Several grammatical/orthographic errors are found throughout the manuscript, including, the following examples:

i. p5: IPO13-/- should be IPO13-/-

ii. The legend to Fig 1D "MA plot showing differential expression genes..." should read "MA plot showing differentially expressed genes..."

Response to Reviewers' Comments

The changes made to manuscript NCOMMS-18-19338B: "Importin 13 plays a key role in the oxidative stress response through nuclear transport of key transcription factors" in response to the comments of the Reviewers are highlighted in red in the Manuscript File with Tracked Changes. All of the specific changes are detailed below:

Reviewer #1

1 The authors tend to interpret the acute toxicity of extremely high concentrations of hydrogen peroxide as oxidative stress. The physiological relevance is questionable.

Hydrogen peroxide (H_2O_2) is in fact a byproduct of normal cellular metabolism as well as a key metabolite in oxidative stress, and accordingly has been extensively utilized to examine oxidative stress in mammalian cell models at concentrations equal to or greater than those used in the present manuscript^{1,3}. To make this clearer, we now spell this out in the manuscript where we first use H_2O_2 in the manuscript (p. 5, line 93-96), together with references, as appropriate. We thank the Reviewer for encouraging us to make this point more strongly.

2. As the treatment of hydrogen peroxide is called oxidative stress (or sometimes just called stress) throughout the manuscript including the title, other conditions triggering production of ROS and/or causing oxidative stress should be tested in Figures 4 and 5 along with providing experimental evidence of oxidative stress.

It is important to understand that the dose of H_2O_2 used in the RNA sequencing experiment and in import/export (Fig. 5) and interaction experiments (Fig. 6) is sublethal, effecting little (< 10 % in mESCs; Fig. 4a) to no (in HeLa; new Sup. Fig 8a,b) cell death. To demonstrate, however, that this concentration of H_2O_2 does indeed induce oxidative stress, we now include new data in Sup. Fig. 2 that document the elevation of intracellular ROS by this sublethal H_2O_2 concentration in both mESC and HeLa cells.

Higher H_2O_2 doses are definitely able to trigger oxidative stress-induced death in mESC (eg. Fig. 4). We now include new data in Sup. Fig. 2a,b that these higher H_2O_2 concentrations also induce an increase in the intracellular levels of ROS. This documents the fact that oxidative stress is being induced, as part of the path to cell death to eliminate irreparably damaged cells from the population. We also clarify this point as appropriate in the text (p. 8, line 195-199), and thank the Reviewer for these suggestions.

Finally, we thank the reviewer for the idea to include alternative approaches to induce oxidative stress. We now include data in the form of new Sup Figs. 6 and 10 which reproduce the experiments in Fig 4 and Fig. 5 but use the oxidant diethyl maleate (DEM, used in other studies to induce oxidative stress^{4,7} as spelled out on p. 10, lines 226-230), which works by depleting cellular glutathione. We were able to show that DEM induces intracellular ROS in mESC and HeLa (new Sup. Fig. 2c-d, f). The new data confirms that IPO13's role is not unique to H_2O_2 -induced oxidative stress, but likely to cellular oxidative stress response generally (p. 10, lines 230-239). We thank the Reviewer again for all of these important suggestions.

3. Since knockdown of SP1 or KLF4 is not equivalent or mimicking the deficiency of IPO13, the approach in Figure 4 and the author's interpretation that SP1 and KLF4 both contribute to IPO13-dependent oxidative stress induced cell death (lines 222-223) sound incorrect or misleading. Thus, the molecular mechanism through which the deficiency of IPO13 makes mESC resistant to the toxicity of hydrogen peroxide is not clear.

4. Along the same lines, (if the author would not agree with the point 3), the reproducibility of Figure 4 should be assessed in other cell types.

We thank the Reviewer for encouraging us to clarify this point. We have amended the manuscript to clarify that knockdown of KLF4 or SP1 results in resistance to H₂O₂-induced cell death that is comparable to the resistance where IPO13 itself is knocked out (p. 9, lines 216-224); this is clear evidence that KLF4 and SP1 are potentially key players in the mechanism by which IPO13 regulates cell survival in the oxidative stress response. Given that IPO13's cellular role is in nuclear transport, the implication is that under stress, transport of KLF4 and SP1 to their respective cellular compartments likely affects cell viability. We confirm this nuclear transport role in Figures 5-6.

5. Rescue experiments are also necessary by the introduction of IPO13 and IPO13-regulated genes back to IPO13 KO cells and assessment of susceptibility to hydrogen peroxide.

We thank the reviewer for suggesting rescue experiments, which are now included in a new figure (Sup. Figure 5), that shows that ectopic expression of IPO13 in the IPO13⁻mESC rescues the sensitivity of mESC to H₂O₂ (p. 9, lines 209-214).

6. IPO13 interacting proteins were previously characterized by Y2H screening (Fatima et al 2-17, Ref #4), in which SP1 or KLF4 was not found as an IPO13 binding protein. So, these are new findings. The approach in this work is also different from the previous work; however, due to the lack of connection between SP1 or KLF4 target genes and the mechanism of susceptibility to toxicity of hydrogen peroxide in conjunction with IPO13, the overall of this work is more descriptive than mechanistic; therefore its contribution to the field may be incremental at most.

The Reviewer cites work from Fatima et al. describing a Y2H screen in testicular germ cells for proteins interacting with the testis-specific N-terminally truncated isoform of IPO13 (T-IPO13), but fails to realise that that this is in no way diminishes the seminal significance and breadth of our present study, which focusses on IPO13 function. Our study uses a stem cell KO model to establish IPO13's key role as a transporter in the transcriptomic oxidative stress response for the first time, and determines the mechanistic basis thereof; interactors of full length IPO13 are identified here using a novel approach based on *de novo* promoter motif analysis on the IPO13-dependent genes responsive to oxidative stress described here for the first time. It is a seminal mechanistic study that documents the connection between IPO13 and the transcriptomic response to oxidative stress, the key roles of SP1 and KLF4 in the transcriptomic response, and the novel finding that under stress, both are export cargoes of IPO13. No IPO has previously been shown to be functional as a transporter in stress, so the work here fills a crucial gap in knowledge in the field of stress biology. To help the reader better understand this, and satisfy the Reviewer, we have reinforced the background (p. 3, lines 55-59, and p. 14, lines 335-350) to put the significance of our contribution in context.

IPO13 enhances the susceptibility of the cells to oxidative stress in part through mediating nuclear export of SP1 and KLF4, which can act as transcriptional activators/repressors with respect to their target genes to effect stress-induced up-/down-regulation to impact cell survival/trigger further transcriptomic responses etc. That SP1/KLF4 target genes are involved in regulation of cell survival is now highlighted in the Results section (p. 8, lines 179-187) and the overall mechanism spelled out more clearly in the Discussion (p. 13, lines 329-341).

Minor

1. Switching cells from mESC to HeLa cells in Fig.5 is not well justified

We now clearly explain the reason for using HeLa (p. 10, lines 243-245) and include new imaging data performed in the mESCs that align with those in HeLas, and confirm IPO13 regulates nuclear export of KLF4 and SP1 in both lines (see new Sup. Figure 9).

2. Line 63: this paragraph is pretty much repetitive of the abstract. The introduction should cover more background and roles of IPO in oxidative stress response tied with the rationale behind this work

We have re-examined the introduction of the manuscript, removed parts that were repetitive of the abstract and included some key background information regarding nuclear transport under conditions of oxidative stress (p. 3, line 55-59).

3. Line 123 about CYP2S1, downregulated in response to what stress? References should be cited here

Line 123 refers to the observations in the present manuscript; accordingly, we now refer (p.6, line 119-121) to the appropriate data (Sup. Table 2 and 3).

4. Line 170 – apoptosis regulation should be clarified as pro- or anti-apoptotic

We have clarified this, as requested (p. 7, lines 159-161).

5. All Western blots need MW markers.

We have included the MW of all bands, as requested for all of Sup. Fig. 4e, Sup. Fig. 7e,j,o, Fig. 5b, Sup. Fig. 9e, Sup. Fig. b,f,i, and Fig. 6a,c .

Reviewer #2

1. The authors claim importin 13 mediates nuclear import of KLF4 and nuclear export of SP1, but images in Figure 5 are unclear (especially, Fig 5a, f + IPO13 untreated). Merging images might help. In addition, already established nuclear import substrate(s) and nuclear export substrate(s) should be shown as a positive control. Also, expression levels of proteins (SP1, KLF4) and over expressed levels of importin 13 should be examined by western blotting.

To address the Reviewer's concerns, the figures specified have been replaced (new Fig. 5a,c and d,e), with new data included to demonstrate the effect of IPO13 on the localisation of ectopically expressed KLF4. As requested, images have been merged where appropriate, and positive control data included for IPO13 import/export cargoes UBC9^s and EIF1A^s, respectively (new Sup. Fig. 7). Finally, we supply Western blots documenting expression levels of SP1 (new Sup. Fig. 10i), KLF4 (new Sup. Fig. 10. f) and IPO13 (new Sup. Fig. 7j,o), as requested. We thank the Reviewer for these suggestions.

2. RanGTP dependency of KLF4 and SP1 binding to importin 13 must be examined

We thank the Reviewer for encouraging us to examine the impact of Ran on KLF4 and SP1 binding to IPO13. We now include new Figure 6, which, apart from showing oxidative stress-enhanced binding of KLF4 or SP1 to IPO13, establishes that SP1 and KLF4 can both bind to IPO13 in the presence of the non-hydrolysable GTP analogue GTP γ S, which maintains Ran in the GTP bound form (p. 13, lines 314-326). Importantly, addition of GTP γ S to the cell lysates in the case of both KLF4 and SP1 appears to enhance binding to IPO13, implying that RanGTP likely enhances the binding.

Reviewer #3

1. A main concern of this reviewer is that the authors do not provide a rationale from mESC to HeLa cells for the IPO13-dependent import/export and interaction studies presented in Figures 5 and 6? What impact on the authors data interpretation and conclusions would a potential difference in the mechanistic and gene-specific role(s) of ipo13 between mES and HeLa cells have? Differences in IPOs and their function between cells and tissues including ESCs and differentiated progenies clearly exist (see for example the review by Belmonte and colleagues (Yang et al.)

Ectopic expression work with ESCs is hampered by very low transfection efficiency, and so for transfection experiments, the commonly used, efficiently transfected HeLa cell line was chosen. We have now detailed this in the manuscript (p.10, lines 243-245), as requested. To allay the reviewer's concerns regarding differences in the cell systems, we now include new Sup. Fig. 9 which shows results consistent with IPO13 mediation of nuclear export of KLF4 and SP1 in both ESCs and HeLa. New Sup Fig. 9a-d shows results for endogenous Sp1 and KLF4, while new Sup Fig 9f-g and Fig. 5a-e document IPO13 export of ectopic KLF4 in both cell lines. We thank the Reviewer for asking us to clarify this.

2. Another concern lies with regard to Suppl Fig 3: This reviewer's understanding of the figure legend "IPO13^{+/+} ESCs were treated without or with siRNA targeting [gene name]" is that untreated IPO13^{+/+} ESCs were compared to siRNA treated ESCs. This understanding seems underscored by the figure itself, which says "IPO13^{+/+} and not SCRAM siRNA vs siRNA [gene name]. This is not an appropriate comparison. Instead IPO13^{+/+} cells treated with the non-targeting (SCRAM siRNA) control pool (D001810-10-50) are the appropriate control, as the authors have done for Fig. 5.

We have amended the figure legend and figure labels in this Figure (now Sup. Fig. 4), as appropriate – we thank the Reviewer for highlighting this point of confusion.

3. Fig 5f and g are referred to before 5d and e, thus the order should be switched in the figure.

We thank the Reviewer; we have amended this as requested.

4. Several grammatical/orthographic errors are found throughout the manuscript, including the following examples: i. p5: IPO13^{-/-} should be IPO13^{-/-} ii. The legend to Fig 1D "MA plot showing differential expression genes..." should read "MA plot showing differentially expressed genes..."

We thank the Reviewer for pointing out these errors, all of which have been corrected, together with a number of other similar errors we have found at the Reviewer's insistence.

We thank the Reviewer again for his/her important contribution to making our manuscript a more rigorous evocation of work interest for publication in *Nature Communications*.

- 1 Yang, K. *et al.* A redox mechanism underlying nucleolar stress sensing by nucleophosmin. *Nature communications* **7**, 13599, doi:10.1038/ncomms13599 <https://www.nature.com/articles/ncomms13599#supplementary-information> (2016).
- 2 Ou, X., Lee, M. R., Huang, X., Messina-Graham, S. & Broxmeyer, H. E. SIRT1 positively regulates autophagy and mitochondria function in embryonic stem cells under oxidative stress. *Stem cells (Dayton, Ohio)* **32**, 1183-1194, doi:10.1002/stem.1641 (2014).
- 3 Guo, Z., Kozlov, S., Lavin, M. F., Person, M. D. & Paull, T. T. ATM activation by oxidative stress. *Science (New York, N.Y.)* **330**, 517-521, doi:10.1126/science.1192912 (2010).
- 4 Faraonio, R. *et al.* A set of miRNAs participates in the cellular senescence program in human diploid fibroblasts. *Cell death and differentiation* **19**, 713-721, doi:10.1038/cdd.2011.143 (2012).
- 5 Kodiha, M. *et al.* Oxidative stress mislocalizes and retains transport factor importin-alpha and nucleoporins Nup153 and Nup88 in nuclei where they generate high

molecular mass complexes. *Biochimica et biophysica acta* **1783**, 405-418, doi:10.1016/j.bbamcr.2007.10.022 (2008).

6 Mahboubi, H., Seganathy, E., Kong, D. & Stochaj, U. Identification of Novel Stress Granule Components That Are Involved in Nuclear Transport. *PloS one* **8**, e68356, doi:10.1371/journal.pone.0068356 (2013).

7 Crampton, N., Kodiha, M., Shrivastava, S., Umar, R. & Stochaj, U. Oxidative stress inhibits nuclear protein export by multiple mechanisms that target FG nucleoporins and Crm1. *Molecular biology of the cell* **20**, 5106-5116, doi:10.1091/mbc.E09-05-0397 (2009).

8 Mingot, J. M., Kostka, S., Kraft, R., Hartmann, E. & Gorlich, D. Importin 13: a novel mediator of nuclear import and export. *The EMBO journal* **20**, 3685-3694, doi:10.1093/emboj/20.14.3685 (2001).

9 Grunwald, M., Lazzaretti, D. & Bono, F. Structural basis for the nuclear export activity of Importin13. *The EMBO journal* **32**, 899-913, doi:10.1038/emboj.2013.29 (2013).

REVIEWER COMMENTS

Reviewer #1 (Remarks to the Author):

This manuscript, originally reviewed in August 2018, was revised in April 2021. Many of the weaknesses were addressed. However, this reviewer was confused by the revised manuscript because of the inconsistent conclusion between the original and revised versions.

Major issue

1. The revised manuscript has the opposite results and conclusion to the original submission in regard to the role of IPO13 in the nuclear transport of KLF4. The original manuscript confirmed and concluded IPO13 as a nuclear importer of KLF4, while in the revised manuscript as a nuclear exporter.

In the original manuscript,

The clear implication is that IPO13 may act as a nuclear importer for KLF4. To confirm this, we tested the effect of overexpression of GFP-IPO13 on KLF4 localisation (Fig. 5f-g), which significantly enhanced nuclear localisation in either the absence or presence of H₂O₂ treatment (The original manuscript lines 234-238).

SP1 differed from KLF4 in that nuclear localisation of SP1 (Fig. 5d-e) was strongly nuclear (Fn/c of c. 10) but this was decreased by oxidative stress (Fn/c of c. 4) or siRNA to IPO13 (Fn/c of c. 7). IPO13 knockdown also reduced the relocalisation of SP1 to the cytoplasm in response to H₂O₂ (Fn/c of c. 7 compared to a value of 4 in the control). The clear implication is that IPO13 may act as a nuclear exporter for SP1. To confirm this, we tested the effect of overexpression of GFP-IPO13 on SP1 localisation (Fig. 5h), with clearly enhanced cytoplasmic accumulation of SP1 in cells whether subjected to oxidative stress or not. Quantitative analysis (Fig. 5i) confirmed the results, with significantly ($p < 0.0001$) reduced nuclear accumulation (lower Fn/c) in the presence of overexpressed IPO13 (The original manuscript lines 240-250).

Taken together, this data shows that IPO13 can both interact with and facilitate nuclear import or export of KLF4 and SP1, respectively. (Lines 257-259)

More major points

1. The effect of KLF4 knockdown agree with the line 172-174 interpretation; however, the knocking down Sp1 or RREB1 per se increased the basal expression of SNAI3, which may have contributed to no further SNAI3 mRNA increase by H₂O₂ treatment. The same issue in IL5 mRNA.

2. CellRox fluorescence intensity in new Fig. S2a,b is not dose-dependent from the concentrations of 125 to 1000 μ M H₂O₂ (corresponding to the lines 195-199).

3. Along the same lines, do IPO13+/+ cells accumulate more ROS than -/- cells (in Fig. S2) or no difference?

4. Frequent referring to supplemental data will make readers disregard them or quite difficult to follow this manuscript. Some of the supplemental data should be moved to the core main figures if the journal restriction still allows to do so, otherwise only Figures 4 to 6 are primary and biological evidence for the conclusion.

Minor

1. The Fig. 1C legend needs better explanation of several genes the authors selected (corresponding to the line 170) and the reason for SNAI3, LEF1, and FOSB in both yellow and purple circles (rather than in the area of 11 overlapping genes).

2. What does the new paragraph (lines 179-187) mean (for which result)? The line 180 "a number are linked to cell survival" is totally puzzling.

3. All supplemental file names are something like 173198_2_data_set_xxxx. It is very difficult to find the data that the authors refer to without figure numbers. Labeling a figure number (including the main figures) would be much appreciated and generally helpful for reviewers.

Reviewer #2 (Remarks to the Author):

The authors have performed several requested experiments properly, and I am satisfied with the revised manuscript. i.e. The authors examined the effect of Ran on KLF4 and SP1 binding to IPO13 in the presence of GTPγS. Addition of GTPγS to the cell lysate is expected to generate Ran GTP that reduces the binding of import substrate (KLF4) to IPO13 while it increases the binding of export substrate (SP1) to IPO13. The results shown in Fig6 is applicable (simple addition of GTPγS to cell lysate is occasionally not so efficient for generating RanGTP, GEF activity is usually required).

Minor comment:

Introduction p2, line 36 "Importin13(IPO13) is one of only two mammalian family members capable of mediating both nuclear import and export".

This sentence should be changed to "Importin13(IPO13) is one of mammalian family members capable of mediating both nuclear import and export".

We now have more example for bidirectional transporter (Importin13, Exportin 4, Exportin 7) , see J Cell Biol. 2018 Jul 2; 217(7): 2329–2340.

Reviewer #3 (Remarks to the Author):

The revised version is greatly improved over the original manuscript. While I disagree with the authors statement to my comment 1 that mESC are difficult to transfect (there is ample documentation on sufficient transfection efficiency and exogenous gene expression or siRNA KD in mESC, moreover, lentiviral transduction is an alternatively way for exogenous gene expression or shRNA KD), I appreciate and accept the authors efforts and additional data presented in Sup Fig. 9. I still raise a few comments below, in no specific ranking order, that need/should be addressed before the manuscript can officially be accepted and move to publication.

1. In Abstract line 22, it should be "...was shown to.." instead of "...could be shown to...".

2. The sentence "Significantly in this context, knock out of IPO13 protects mESCs from oxidative stress-induced death, with siRNA knockdown of either SP1 or KLF4 able to confer resistance to H2O2." (lines 68-70) is slightly confusing. SP1 KD or KLF4 KD in IPO13+/+ cells protect from H2O2 induced stress, similar to what is seen in "just" IPO13-/- cells treated with H2O2. It is not SP1 KD or KLF4 KD in IPO13-/- cells, as a reader may interpret the sentence. Please clarify/rectify.

3. Regarding Sup. Fig. 4, what was the authors' rationale for not testing all genes under all conditions, or, in other words, how did the authors decide which gene to test under which condition? Also, the figure is slightly confusing in its organization insofar as that tested genes appear on different rows, which makes it difficult to readily see the effect of the various KDs on that gene. Finally, p-values should be given for the comparison of NT siRNA vs "gene" siRNA, at least for the H2O2 condition, as this is the significance the authors are actually interested in and describing in the text.

4. Lines 181-187 make one extremely long sentence, describing STOX1, PPARδ, IL5 and RASD1. The content would be easier to understand if this one sentence were broken up into a few sentences.

5. Although a minor point, I don't follow why was the concentration of H₂O₂ was raised to 1000 uM in Sup Fig 5 instead of the 600 uM used in Fig 4.

6. Lines 254 and 266-267: What is the meaning of "c" in the 5 instances of "Fn/C of c. [number]"? "c" was not included in "Fn/c of 37 and 19 respectively" (lines 261/262) nor was it used/described in the methods section (lines 496-502).

7. Lines 283/284: I assume that "In contrast, SP1 remained highly nuclear with or without H₂O₂ treatment (Fn/c 4.6 and 4.3 respectively)." is describing the data in IPO13^{-/-} cells. The authors need to clarify this.

8. Line 296 "...were immunostained previously described." should be "...were immunostained as previously described."

9. Sup Fig 3: It appears that the symbol labels are wrong or, at least, confusing in the figure itself. The figures show blue and red circles on the same horizontal line with the word "untreated", while below are green and purple triangles with the word "+H₂O₂". However in the various panels, blue and red circles are associated with IPO13^{+/+} cells and green and purple triangles with IPO13^{-/-} cells. Please clarify/rectify.

Reviewer #1 (Remarks to the Author):

This manuscript, originally reviewed in August 2018, was revised in April 2021. Many of the weaknesses were addressed. However, this reviewer was confused by the revised manuscript because of the inconsistent conclusion between the original and revised versions.

Major issue

1. The revised manuscript has the opposite results and conclusion to the original submission in regard to the role of IPO13 in the nuclear transport of KLF4. The original manuscript confirmed and concluded IPO13 as a nuclear importer of KLF4, while in the revised manuscript as a nuclear exporter.

In the original manuscript,

The clear implication is that IPO13 may act as a nuclear importer for KLF4. To confirm this, we tested the effect of overexpression of GFP-IPO13 on KLF4 localisation (Fig. 5f-g), which significantly enhanced nuclear localisation in either the absence or presence of H₂O₂ treatment (The original manuscript lines 234-238).

SP1 differed from KLF4 in that nuclear localisation of SP1 (Fig. 5d-e) was strongly nuclear (Fn/c of c. 10) but this was decreased by oxidative stress (Fn/c of c. 4) or siRNA to IPO13 (Fn/c of c. 7). IPO13 knockdown also reduced the relocalisation of SP1 to the cytoplasm in response to H₂O₂ (Fn/c of c. 7 compared to a value of 4 in the control). The clear implication is that IPO13 may act as a nuclear exporter for SP1. To confirm this, we tested the effect of overexpression of GFP-IPO13 on SP1 localisation (Fig. 5h), with clearly enhanced cytoplasmic accumulation of SP1 in cells whether subjected to oxidative stress or not. Quantitative analysis (Fig. 5i) confirmed the results, with significantly ($p < 0.0001$) reduced nuclear accumulation (lower Fn/c) in the presence of overexpressed IPO13 (The original manuscript lines 240-250).

Taken together, this data shows that IPO13 can both interact with and facilitate nuclear import or export of KLF4 and SP1, respectively. (Lines 257-259)

In the original submission, the experiments examining the effect of GFP-IPO13 overexpression or IPO13 knockdown on KLF4 localisation that led us to conclude IPO13 could facilitate nuclear import of KLF4 were based on immunofluorescence (IF) in HeLa cells using a commercial (abcam) antibody that was not tested/verified by abcam for IF. In addressing the Reviewers' comments in the previous revision, we performed IF experiments in mESCs using a different KLF4 antibody (abcam) that has been tested/verified by abcam for IF, and cited in many publications, including high profile papers in *Nature Medicine*, *Nature Cell Death & Differentiation* and *Nature Oncogene*. We found that H₂O₂-induced stress increased KLF4 cytoplasmic localisation in WT but not KO mESCs, indicating IPO13 likely exports KLF4 in response to stress (Sup. Fig. 8). We then confirmed the findings with new HeLa cell experiments looking at the effect of IPO13 overexpression/knockdown on GFP-KLF4 localisation (Fig. 5b-d); results were identical to those in mESCs, with H₂O₂-induced stress triggering cytoplasmic localisation of KLF4 which could be enhanced or reduced by IPO13 overexpression or knockdown respectively, supporting the nuclear export role for IPO13. It seems reasonable to conclude that the misleading original "nuclear import" results can be attributed to the unverified antibody. Our new in-depth analysis using the verified antibody shows results consistent across mESCs and HeLa cells, supporting our revised conclusion as to IPO13's transport role.

More major points

1. The effect of KLF4 knockdown agree with the line 172-174 interpretation; however, the knocking down Sp1 or RREB1 per se increased the basal expression of SNAI3, which may have contributed to no further SNAI3 mRNA increase by H₂O₂ treatment. The same issue in IL5 mRNA.

We thank the Reviewer for the insightful comments, which have encouraged us to perform our analysis in a more rigorous way; we are very pleased with the outcome (new Supp. Fig. 3a-d) ! With respect to *Il5* and to *Snai3* where SP1 is knocked down, we determined that there is no significant difference in basal expression (p values 0.1044 and 0.3767 for *Il5* without or with KLF4 or NFAT1 knockdown respectively, and p value of 0.2443 for *Snai3* without and with SP1 knock down); we thank the Reviewer for encouraging us to investigate this possibility. In the case of *Snai3* with and without KLF4 or RREB1 siRNA, there was a significant difference (p values of 0.0003 and 0.0144 respectively) – again, we thank the Reviewer; we conclude this is indicative of RREB1 and KLF4 being involved in *Snai3* basal gene expression as well as in stress, and now point this out in the Results section (Lines 172-179). To correct for such differences in basal expression, we have relativised to basal levels for respective unstressed cells in all our revised analysis for Fig. 3a-d; this reveals that the stress response of the majority of genes analysed (but not the *Cyp21* control) is abrogated by knockdown of at least one of the four TFs. We are

indebted to the Reviewer for encouraging us to analyse our important results in a more rigorous way, that gives us confidence in the overall results and their interpretation.

2. CellRox fluorescence intensity in new Fig. S2a,b is not dose-dependent from the concentrations of 125 to 1000 μM H_2O_2 (corresponding to the lines 195-199).

Previous studies have found that H_2O_2 concentrations can induce a dose-dependent increase in CellRox fluorescence, but then plateau or decrease at concentrations approaching the upper limit of sublethality, which may be indicative of signal saturation and/or oxidative stress inducing membrane damage leading to CellRox dye leakage¹. Consistent with this idea, the sublethal concentration of 125 μM we use in our study is where the signal for CellRox plateaus. Therefore, we now include data for ROS accumulation measured immediately after treatment with 125 μM H_2O_2 and measured after allowing cells to recover for 2h post treatment with 125 μM H_2O_2 . We show a significant increase in ROS accumulation/CellRox fluorescence in 125 μM H_2O_2 treated cells and further, that this fluorescence is then reduced, returning to basal level, when the media is replaced and cells are allowed to recover from the stress (Figure 1 e-f). We feel this data better captures the relevant treatments of our study in this respect, and thank the Reviewer for encouraging us to present our analysis in this way.

3. Along the same lines, do IPO13^{+/+} cells accumulate more ROS than -/- cells (in Fig. S2) or no difference?

Comparing IPO13^{+/+} and IPO13^{-/-} cells, there is a trend toward the IPO13^{-/-} cells accumulating less ROS when treated with 125 μM H_2O_2 , but this difference is not statistically significant (now Fig. 1e-f).

4. Frequent referring to supplemental data will make readers disregard them or quite difficult to follow this manuscript. Some of the supplemental data should be moved to the core main figures if the journal restriction still allows to do so, otherwise only Figures 4 to 6 are primary and biological evidence for the conclusion.

We recognise that the supplementary data in our manuscript has grown considerably since the original submission, and that this this makes it a bit more challenging for readers to follow. We thank the Reviewer for suggesting that we move some supplemental data to the core main figures. We have now taken out Sup. Fig 1. and included the data from the figure across three other figures in the main paper, including Figure 1, where the data is inserted into panels e and f. We thank the Reviewer for encouraging us to do this.

Minor

1. The Fig. 1C legend needs better explanation of several genes the authors selected (corresponding to the line 170) and the reason for SNAI3, LEF1, and FOSB in both yellow and purple circles (rather than in the area of 11 overlapping genes).

We thank the Reviewer for making us rethink presentation of the information in the Venn diagram in Fig. 3c and corresponding legend. To address this, we have redrawn the Venn diagram to firstly be a more accurate area-proportional representation of the data in terms of number of genes in all parts of the diagram, and annotated RT-qPCR gene targets precisely in the appropriate portions of the Venn diagram. For example, the genes *Snai3* and *Fosb* are located in the portion where the SP1/KLF4 motif and the SP1/RREB-1 motif overlap, since the motifs that aligned to these TFs was found within both of these genes. The motif aligned to the NFAT1 binding motif was not found in *Snai3* or *Fosb*. All three motifs were identified in *Lef1* and thus *Lef1* is found in the portion overlapping the three motifs. Importantly, this has now been clarified in the Fig 3c legend (Lines 856-864). We thank the Reviewer for helping make our presentation clearer, and thereby increase the impact of the work.

2. What does the new paragraph (lines 179-187) mean (for which result)? The line 180 "a number are linked to cell survival" is totally puzzling.

Apologies for the confusion - the paragraph in question is not describing results in the manuscript, but referring to the published literature with respect to the genes identified in our study as IPO13-dependent/dependent on the respective TF having reported roles in regulation of ROS homeostasis and

cell death. To avoid confusion, we now clarify which findings are from the present study, and the literature (Refs 63-69; see lines 191-193).

3. All supplemental file names are something like 173198_2_data_set_xxxx. It is very difficult to find the data that the authors refer to without figure numbers. Labeling a figure number (including the main figures) would be much appreciated and generally helpful for reviewers.

We apologize for any inconvenience the Reviewer experienced here. Although we were careful to label each file name with the respective figure number, we now also make sure that all figure numbers are clearly visible within each figure file.

Reviewer #2 (Remarks to the Author):

The authors have performed several requested experiments properly, and I am satisfied with the revised manuscript. i.e. The authors examined the effect of Ran on KLF4 and SP1 binding to IPO13 in the presence of GTPγS. Addition of GTPγS to the cell lysate is expected to generate Ran GTP that reduces the binding of import substrate (KLF4) to IPO13 while it increases the binding of export substrate (SP1) to IPO13. The results shown in Fig6 is applicable (simple addition of GTPγS to cell lysate is occasionally not so efficient for generating RanGTP, GEF activity is usually required).

We would like to thank the Reviewer once again, for his/her valuable suggestions that have helped to improve our work in the process of review.

Minor comment:

Introduction p2, line 36 "Importin13(IPO13) is one of only two mammalian family members capable of mediating both nuclear import and export".

This sentence should be changed to "Importin13(IPO13) is one of mammalian family members capable of mediating both nuclear import and export".

We now have more example for bidirectional transporter (Importin13, Exportin 4, Exportin 7) , see J Cell Biol. 2018 Jul 2; 217(7): 2329–2340.

We thank the reviewer for bringing this outdated statement to our attention, we have updated the statement to match the current literature on this topic, including the recommended reference (Lines 36-39; new Ref. 5).

Reviewer #3 (Remarks to the Author):

The revised version is greatly improved over the original manuscript. While I disagree with the authors statement to my comment 1 that mESC are difficult to transfect (there is ample documentation on sufficient transfection efficiency and exogenous gene expression or siRNA KD in mESC, moreover, lentiviral transduction is an alternatively way for exogenous gene expression or shRNA KD), I appreciate and accept the authors efforts and additional data presented in Sup Fig. 9. I still raise a few comments below, in no specific ranking order, that need/should be addressed before the manuscript can officially be accepted and move to publication.

1. In Abstract line 22, it should be "...was shown to.." instead of "...could be shown to...".

We have made this amendment to the abstract.

2. The sentence "Significantly in this context, knock out of IPO13 protects mESCs from oxidative stress-induced death, with siRNA knockdown of either SP1 or KLF4 able to confer resistance to H₂O₂." (lines 68-70) is slightly confusing. SP1 KD or KLF4 KD in IPO13^{+/+} cells protect from H₂O₂ induced stress, similar to what is seen in "just" IPO13^{-/-} cells treated with H₂O₂. It is not SP1 KD or KLF4 KD in IPO13^{-/-} cells, as a reader may interpret the sentence. Please clarify/rectify.

We thank the Reviewer for highlighting the confusion. We have amended the sentence to clarify that it is in fact knock down of SP1 or KLF4 in the IPO13^{+/+} cells that confers resistance to H₂O₂ rather than in the IPO13^{-/-} cells.

3. Regarding Sup. Fig. 4, what was the authors' rationale for not testing all genes under all conditions, or, in other words, how did the authors decide which gene to test under which condition? Also, the figure is slightly confusing in its organization insofar as that tested genes appear on different rows, which makes it difficult to readily see the effect of the various KDs on that gene. Finally, p-values should

be given for the comparison of NT siRNA vs "gene" siRNA, at least for the H₂O₂ condition, as this is the significance the authors are actually interested in and describing in the text.

To confirm that the transcription factors SP1, KLF4, NFAT1 and RREB-1 regulate the transcription of IPO13-dependent stress responsive genes, within which the promoter was found to contain the respective transcription factor binding motif, we performed qPCR analysis on selected genes. Genes were tested via TF knockdown only if the motif was identified in their promoter. For example, *Snai3* was found to contain the motif aligned to transcription factors KLF4, SP1 and RREB-1, but not NFAT1. We also tested the effect of knockdown of each of the TFs on the IPO13-independent gene *Cyp2s1* as a negative control, finding that knockdown of any of the TF did not affect its stress induced downregulation. We recognise that the way the genes are organised in Sup Fig. 4 (now Sup. Fig. 3) made it difficult to compare the effect of each TF on a single gene. We thank the Reviewer for pointing this out, and we have now rearranged the panels to improve on this point. Finally, as per the suggestions of Reviewer 1 (major point 1), we have now relativised all results to basal for the respective non-targeting and targeted siRNA-treated cells and highlight the significant differences, all in Sup Fig. 3 – we thank the Reviewer.

4. Lines 181-187 make one extremely long sentence, describing STOX1, PPAR δ , IL5 and RASD1. The content would be easier to understand if this one sentence were broken up into a few sentences.

We have broken the sentence down into a few sentences as suggested by the reviewer.

5. Although a minor point, I don't follow why was the concentration of H₂O₂ was raised to 1000 μ M in Sup Fig 5 instead of the 600 μ M used in Fig 4.

We agree with the Reviewer that it is a minor point - Fig 4 and Sup. Fig 5 (now Sup. Fig. 4) are different types of experiments and hence not meant to be directly compared with respect to stress treatment, since the cells are subjected to rather different manipulations:

- In Fig 4, cells are seeded and transfected with siRNA (Dharmafect I transfection reagent) three days prior to treatment with H₂O₂

- In Sup. Fig. 4, cells are seeded and transfected (Lipofectamine 2000 transfection reagent) to express GFP/GFP-tagged proteins one day before treatment with H₂O₂

6. Lines 254 and 266-267: What is the meaning of "c" in the 5 instances of "Fn/C of c. [number]"? "c" was not included in "Fn/c of 37 and 19 respectively" (lines 261/262) nor was it used/described in the methods section (lines 496-502).

The term circa used to indicate approximately, has been abbreviated to "c." here.

7. Lines 283/284: I assume that "In contrast, SP1 remained highly nuclear with or without H₂O₂ treatment (Fn/c 4.6 and 4.3 respectively)." is describing the data in IPO13^{-/-} cells. The authors need to clarify this.

We thank the Reviewer, and have amended the statement to clarify which comparison is being discussed here.

8. Line 296 "...were immunostained previously described." should be "...were immunostained as previously described.".

We thank the Reviewer; we have amended this.

9. Sup Fig 3: It appears that the symbol labels are wrong or, at least, confusing in the figure itself. The figures show blue and red circles on the same horizontal line with the word "untreated", while below are green and purple triangles with the word "+H₂O₂". However in the various panels, blue and red circles are associated with IPO13^{+/+} cells and green and purple triangles with IPO13^{-/-} cells. Please clarify/rectify.

We thank the Reviewer. We have amended this figure, now Sup. Fig 2, which now shows that blocked out blue circles and blocked out green circles are untreated samples in IPO13^{+/+} and IPO13^{-/-} cells respectively. Empty circles and triangles denote H₂O₂ treated IPO13^{+/+} and IPO13^{-/-} cells respectively.

References:

¹ Daniel Manoil, Serge Bouillaguet. Oxidative Stress in Bacteria Measured by Flow Cytometry. *Adv Biotech & Micro.* 2018; 8(1): 555726. DOI: 10.19080/AIBM.2018.08.555726

REVIEWERS' COMMENTS

Reviewer #2 (Remarks to the Author):

I am satisfied with the authors' revision.